# SARS-CoV-2 induces robust germinal center CD4 T follicular helper cell responses in rhesus macaques

Yashavanth Shaan Lakshmanappa [1,14], Sonny R. Elizaldi[1,2,14], Jamin W. Roh[1,2,14], Brian A. Schmidt[1,14], Timothy D. Carroll[3,14], Kourtney D. Weaver[4], Justin C. Smith [4], Anil Verma[1], Jesse D. Deere[3], Joseph Dutra[3], Mars Stone[5,6], Sergej Franz[6], Rebecca Lee Sammak[3], Katherine J. Olstad[3], J. Rachel Reader [3,7], Zhong-Min Ma [3], Nancy K. Nguyen[1], Jennifer Watanabe [3], Jodie Usachenko[3], Ramya Immareddy[3], JoAnn L. Yee [3], Daniela Weiskopf[8], Alessandro Sette[8,9], Dennis Hartigan-O'Connor[3,10], Stephen J. McSorley[1,11], John H. Morrison[3,12], Nam K. Tran[13], Graham Simmons[5,6], Michael P. Busch[5,6], Pamela A. Kozlowski [4], Koen K. A. Van Rompay [3,7✉], Christopher J. Miller [1,3,7✉] & Smita S. Iyer [1,3,7✉]

CD4 T follicular helper ($T_{fh}$) cells are important for the generation of durable and specific humoral protection against viral infections. The degree to which SARS-CoV-2 infection generates $T_{fh}$ cells and stimulates the germinal center (GC) response is an important question as we investigate vaccine induced immunity against COVID-19. Here, we report that SARS-CoV-2 infection in rhesus macaques, either infused with convalescent plasma, normal plasma, or receiving no infusion, resulted in transient accumulation of pro-inflammatory monocytes and proliferating $T_{fh}$ cells with a $T_h1$ profile in peripheral blood. CD4 helper cell responses skewed predominantly toward a $T_h1$ response in blood, lung, and lymph nodes. SARS-CoV-2 Infection induced GC $T_{fh}$ cells specific for the SARS-CoV-2 spike and nucleocapsid proteins, and a corresponding early appearance of antiviral serum IgG antibodies. Collectively, the data show induction of GC responses in a rhesus model of mild COVID-19.

[1] Center for Immunology and Infectious Diseases, UC Davis, Davis, CA, USA. [2] Graduate Group in Immunology, UC Davis, Davis, CA, USA. [3] California National Primate Research Center, UC Davis, Davis, CA, USA. [4] Department of Microbiology, Immunology, and Parasitology, Louisiana State University Health Sciences Center, New Orleans, LA, USA. [5] Department of Laboratory Medicine, University of California, San Francisco, CA, USA. [6] Vitalant Research Institute, San Francisco, CA, USA. [7] Department of Pathology, Microbiology, and Immunology, School of Veterinary Medicine, UC Davis, Davis, CA, USA. [8] Center for Infectious Disease and Vaccine Research, La Jolla Institute for Immunology, La Jolla, San Diego, CA, USA. [9] Department of Medicine, Division of Infectious Diseases and Global Public Health, University of California, La Jolla, San Diego, CA, USA. [10] Department of Medical Microbiology and Immunology, School of Medicine, UC Davis, Davis, CA, USA. [11] Department of Anatomy, Physiology, and Cell Biology, School of Veterinary Medicine, UC Davis, Davis, CA, USA. [12] Department of Neurology, School of Medicine, UC Davis, Davis, CA, USA. [13] Pathology and Laboratory Medicine, School of Medicine, UC Davis, Davis, CA, USA. [14] These authors contributed equally: Yashavanth Shaan Lakshmanappa, Sonny R. Elizaldi, Jamin W. Roh, Brian A. Schmidt, Timothy D. Carroll. ✉email: kkvanrompay@ucdavis.edu; cjmiller@ucdavis.edu; smiyer@ucdavis.edu

Since the declaration of COVID-19 as a pandemic on 11 March 2020, the virus has rapidly disseminated globally resulting in >50 million infections and more than a million deaths[1]. Although SARS-CoV-2 causes mild or asymptomatic disease in most cases[2], unanticipated post-infection complications, such as multisystem inflammatory syndrome pose a serious threat[3]. An effective vaccine is paramount, and several SARS-CoV-2 vaccine candidates are in various phases of human testing worldwide[4–6]. The most-effective vaccines induce antibodies that provide long-term protection[7] and vaccines using attenuated virus elicit the most persistent antibody responses; therefore, understanding the immunological mechanisms characteristic of controlled SARS-CoV-2 infection is foundational to the selection of a vaccine capable of abating the pandemic[8,9].

Generation of persistent immunity hinges on CD4 T follicular helper cells ($T_{fh}$). $T_{fh}$ cells are a subset of CD4 helper cells with the specialized ability to migrate to germinal centers (GC) structures populated by highly proliferating GC B cells. GC B cells rely on $T_{fh}$ cells for co-stimulation and cytokines in order to survive, proliferate, and differentiate into memory B cells or plasma cells, critical facets of protective immunity[10]. We and others have demonstrated that peripheral CD4 $T_{fh}$ cells predict antibody durability in the context of HIV and influenza vaccines[11–13]. Although studies in humans demonstrate induction of $T_{fh}$ cells in COVID-19 patients[14–16], the impact of SARS-CoV-2 infection on the generation of GC $T_{fh}$ cells is currently unknown. This is a detrimental gap in knowledge as understanding early correlates of durable antibodies, specifically those that circulate in peripheral blood, will aid in the ultimate selection of effective vaccine candidates. SARS-CoV-2-specific CD4 T cells responding to spike proteins have been observed in the peripheral blood of recovered patients[17,18]. Similar observations have been made with the 2002 SARS-CoV virus[19,20], and studies in mouse models have demonstrated a critical role for CD4 T cells in viral clearance[21]. Together, these data emphasize the need to understand CD4 $T_{fh}$ responses following SARS-CoV-2 infection.

Because healthy rhesus macaques infected with SARS-CoV-2 resist immediate re-challenge with the virus[22–24], we hypothesized that understanding the CD4 $T_{fh}$ and GC response following exposure to SARS-CoV-2 will provide a framework for understanding immune mechanisms of protection. We tested this hypothesis in the setting of a study designed to examine the therapeutic efficacy of convalescent plasma infusion in curbing a nascent infection. Although plasma infusion did not impact viral loads, we report that following infection with SARS-CoV-2, adult rhesus macaques exhibited transient accumulation of activated, proliferating $T_{fh}$ cells with a $T_h1$ profile in their peripheral blood. Perhaps more-pertinent to SARS-CoV-2 as a respiratory virus, infection elicited robust GCs with SARS-CoV-2- reactive $T_{fh}$ cells within the mediastinal lymph nodes. The comprehensive immune analysis in a controlled animal model of mild diseases adds to our understanding of immune responses to SARS-CoV-2.

## Results

**Experimental design and convalescent plasma infusion.** To achieve our primary objective of assessing whether SARS-CoV-2 elicits $T_{fh}$ cells and GC responses, we challenged eight adult rhesus macaques (4–5 y, Table S2) with a high-dose of SARS-CoV-2 ($2 \times 10^6$ PFU). We arrived at a sample size of eight for immunological analysis based on viral kinetics described in young adult rhesus macaques[23]. Virus was administered via the intranasal, intratracheal, and ocular routes. Infection was monitored by following the quantity of viral RNA (vRNA) in nasal washes using quantitative real-time polymerase chain reaction (qRT-PCR). Of the eight infected animals: four did not receive plasma

infusion (Infected), two animals were infused with COVID-19 convalescent human plasma 24 hours following inoculation (I+CP), and two animals were infused with an identical volume of normal plasma lacking antibodies to SARS-CoV-2 (I+NP) (Fig. 1A). We assayed monkey sera for infused antibody using reagents specific for human IgG and found that CP infusion resulted in a clear bolstering of binding antibody titers against spike and nucleocapsid proteins of SARS-CoV-2 (Fig. 1B). However, while pooled CP demonstrated a neutralizing titer of 1:1,149; owing to the estimated 50-fold dilution after infusion (based on 4 ml/kg infusion volume and ~20% extracellular fluid), neutralizing activity in macaque sera 24 hours post infusion fell below the limit of detection (1:40) of the neutralization assay (Fig. 1C). Measurement of binding antibodies against SARS-CoV-2 in pooled convalescent plasma revealed concentrations of anti-S1-IgG at 24.5 μg/ml providing a lower limit of anti-S1 IgG for SARS-CoV-2 neutralization (Fig. 1C). Consistent with lack of neutralizing activity, CP administration did not blunt acute viral loads and high vRNA levels were observed in all animals (Fig. 1D). Histopathological lesions of the lungs confirmed multifocal to locally extensive interstitial pneumonia of mild to moderate severity in infected animals (Figure S1A). However, these histological changes were not accompanied by fever, weight loss, or any other signs of clinical disease (Figure S1B). None of the animals developed acute respiratory distress. In summary, infection of healthy adult rhesus macaques with SARS-CoV-2 resulted in high viremia but generally produced no overt signs of clinical illness, providing a framework to investigate development of protective immune responses.

**SARS-CoV-2 infection leads to a rapid and transient shift in innate immune responses and increases the number of CD4 T follicular helper cells in peripheral blood.** Evaluation of innate immune cell subsets in the peripheral blood (Fig. 2A) revealed no significant changes in either the proportion or absolute counts of neutrophils over time (Fig. 2B). However, rapid and divergent changes in specific myeloid cell subsets were observed. Although CD14+ CD16+ pro-inflammatory monocytes significantly increased at Day 2 with a corresponding increase in CX3CR1 expression (Figure S1C), pDCs decreased in peripheral blood. We also noted a significant increase in myeloid DCs (mDC) within the infected group. Pro-inflammatory chemokines monocyte chemoattractant protein (MCP-1), interferon γ-induced protein-10 (IP-10), interferon-inducible T cell α chemoattractant (I-TAC) were significantly elevated at Day 2 and returned to baseline levels soon thereafter (Fig. 2C). We did not observe significant elevations in pro-inflammatory cytokines interleukin (IL)-6. We observed a direct relationship between serum MCP-1 levels and pro-inflammatory monocytes over the course of infection, whereas pDCs and neutrophil frequencies were inversely related to I-TAC and IL-8 levels, respectively (Fig. 2D). Although no statistically significant changes occurred with IL-6 or IL-10, both cytokines were correlated over the course of infection. The brisk and transient innate immune dynamics following exposure to SARS-CoV-2 are consistent with the minimal changes observed in body weight and oxygen saturation levels and mild overall disease pathology. The observed lack of increase in levels of systemic IL-6, which is associated with severe COVID-19[25], may underlie the asymptomatic/mild illness observed in our animals.

To assess the increase in CD4 $T_{fh}$ cells attributable to SARS-CoV-2, we profiled peripheral blood to capture effector T-cell responses. No evidence of general lymphopenia was observed (Figure S1D). Frequency and absolute counts of activated CXCR5+ CD4 $T_{fh}$ cells, identified by co-expression of Ki-67 and PD-1, significantly increased in all animals at Day 7 regardless of plasma

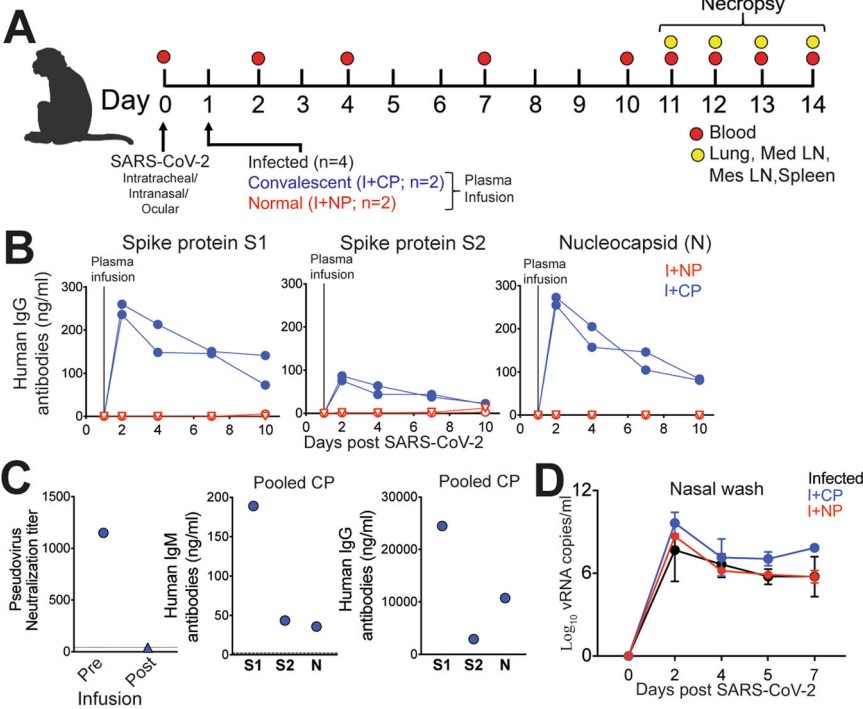

**Fig. 1 Experimental design and convalescent plasma infusion. A** Rhesus macaques inoculated with severe acute respiratory syndrome coronavirus 2 (SARS-CoV-2) were infused with COVID-19 convalescent plasma (I + CP), normal plasma (I + NP), or did not receive plasma (infected). **B** Concentrations of human IgG antibodies against S1, S2, and N following CP infusion. **C** Concentrations of human IgM and IgG antibodies against S1, S2, and N in pooled CP pre-infusion. Pseudovirus neutralization of CP pre-infusion and pooled macaque sera post infusion. **D** Mean viral RNA (+SD) in nasal washes (kinetic data shown derived from independent animals, sample size for experimental groups: $n = 4$ infected, $n = 2$ I + CP, and $n = 2$ I + NP).

intervention (Fig. 3A, B). At the apex of the effector response, Ki-67+ CD4 T cells, specifically the $T_h1$ but not the $T_{fh}$ subset was strongly associated with proliferating CD8 T cells (Fig. 3C). In turn, we observed strong antigen-dependent induction of CD8 T cells evidenced by the association between SARS-CoV-2 vRNA from nasal washes and proliferating (Ki-67+) CD8 T cells.

Evaluation of infection-induced changes in CD4 T-cell differentiation at Day 7 revealed a strong phenotypic shift to $T_h1$ effectors (CXCR3+), $T_h1$ polarized $T_{fh}$ cells (CXCR3+ CXCR5+) and $T_h1$ $T_h17$ (CXCR3+ CCR6+) CD4 T cells as demonstrated by tSNE plots constructed using flow data (Fig. 3D, Figure S2A). Correspondingly, the data showed accumulation of CD4 $T_h1$ cells at Day 7 (Figure S2B). Although $T_h2$ CD4 cells did not peak at Day 7, there was an increase of $T_h17$ CD4 cells (Figure S2C, D). Using the acute activation marker, inducible costimulator (ICOS), to identify proliferating (Ki-67+) CD4 T cells at Day 7 (Fig. 3E), we found that ICOS+CXCR5− and CXCR5+ CD4 T cells subsets expressed the $T_h1$ marker signaling lymphocyte adhesion molecule (SLAM) induced upon T-cell activation[26], which we have shown is also expressed by $T_h1$ $T_{fh}$ cells[13] following immunization, and the fractalkine receptor CX3CR1, a $T_h1$ marker, consistent with their activation status (Fig. 3F). Despite the increase in activated $T_{fh}$ cells, levels of CXCL13 did not increase significantly following SARS-CoV-2 infection (Figure S2E). Neither the ICOS+CXCR5− nor the CXCR5+ CD4 T-cell subsets downregulated CD28 and both subsets expressed CCR7 at levels comparable to or greater than naive cells, indicative of a lymph node-homing phenotype. To assess CD4 T-cell functionality, cytokine production was evaluated ex vivo following stimulation with PMA and ionomycin. Two distinct CD4 T cells were identified—a degranulating CD107a + b subset with the majority of degranulating CD4 T cells expressing interferon gamma (IFNγ) and TNFa but not

IL-2 or IL-17; and an IL-21-producing subset (Figure S2F). In contrast, the majority of IL-21-expressing cells produced IL-2, IL-17, and co-produced TNFa and IFNγ. Thus, CD4 T-cell polyfunctionality was preserved during SARS-CoV-2 infection.

**CD4 $T_{fh}$ cells targeting the spike (S) and nucleocapsid (N) proteins are generated following SARS-CoV-2 infection.** Based on the significant increase in systemic CD4 $T_{fh}$ cells following SARS-CoV-2 exposure, we sought to understand splenic involvement during the GC phase of the immune response. To this end, we quantified GC $T_{fh}$ cells in the spleen at necropsy and compared the values to those seen in splenocytes from opportunistic necropsies in animals that had not been exposed to SARS-CoV-2. The results suggested the initiation of a GC response within the spleen following infection (Figure S3A). We observed that the majority of the GC $T_{fh}$ cells did not express Foxp3 indicating that GC $T_{fh}$ cells predominated over the GC T follicular regulatory cell ($T_{fr}$; CXCR5+, PD-1++, Foxp3+) subset (Figure S3B, Tregs were defined as CD95+ CXCR5− Foxp3+). To conclusively assess SARS-CoV-2-induced responses, we stimulated cryopreserved splenocytes with megapools—overlapping peptides covering multiple T-cell epitopes in S, N, and membrane (M) proteins, and spanning the open reading frames (ORF1,3a,8) of SARS-CoV-2. PMA/Ionomycin was used as a positive control while dimethyl sulfoxide (DMSO)-treated cells served as negative controls. Using activation-induced marker (AIM) assay[27], SARS-CoV-2-specific CD4 T cells were identified based on co-expression of OX40 and CD25 (Fig. 4A). Following subtraction of AIM+ responses in DMSO-treated cells, CD4 T-cell responses to S and N were detected. Furthermore, PD-1++ GC $T_{fh}$ cells, reactive to S, N, and M were observed indicative of SARS-CoV-2-induced GC response in the spleen (Fig. 4B).

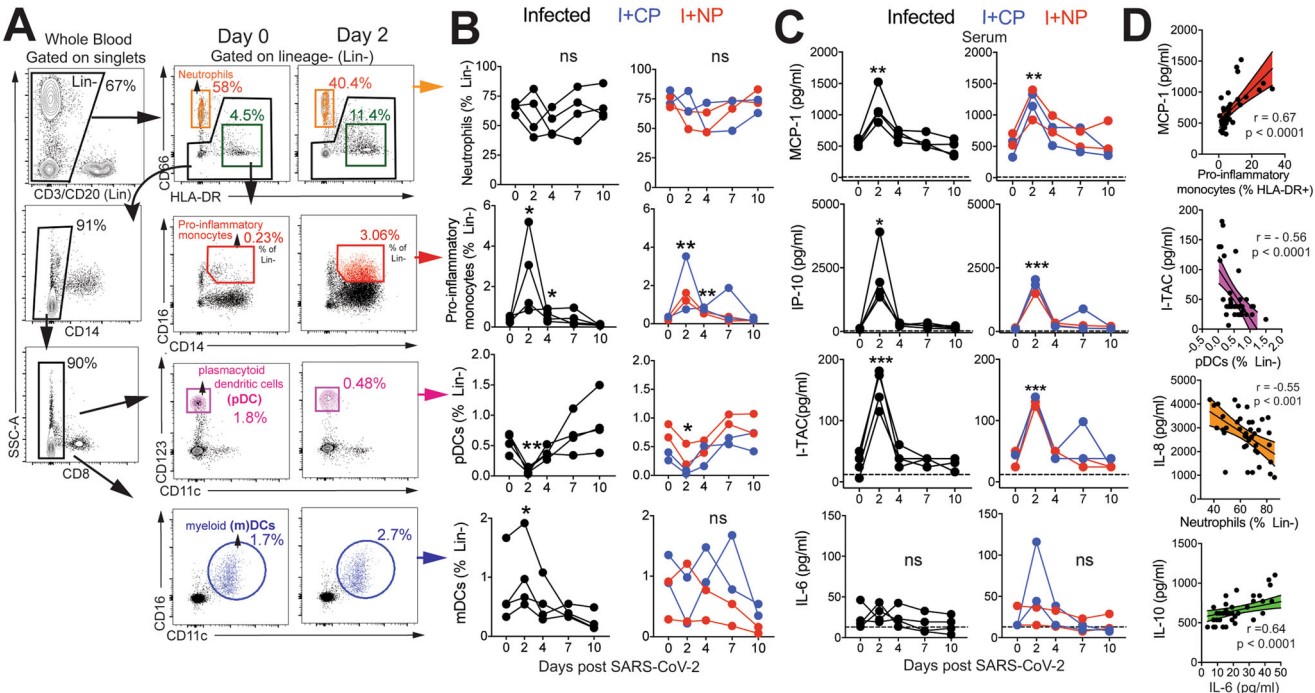

**Fig. 2 SARS-CoV-2 infection leads to rapid and transient shifts in innate immune responses in peripheral blood. A** Representative gating strategy for innate immune subsets in whole blood after gating on singlets. Fluorochromes used were CD3/CD20- APC-Cy7, CD14-A700, CD8- BUV 805, CD66-APC, HLA-DR-BV786, CD16-BV605, CD123-BV421, CD11c-Pe-Cy7. **B** Kinetics of innate immune responses (pro-inflammatory monocytes; *p = 0.01 at d2 and d4 relative to d0 using a one-tailed paired t test in infected animals, **p = 0.006 and 0.002 at d2 and d4 relative to d0 in infused animals, pDCs; **p = 0.005 at d2 relative to d0 using a one-tailed paired t test in infected animals, *p = 0.01 at d2 relative to d0 using a one-tailed paired t test in infused animals, mDCs; *p = 0.02 at d2 relative to d0 using a one-tailed paired t test in infected animals). **C** Serum chemokines monocyte chemoattractant protein (MCP)-1, interferon gamma induced protein (IP)-10, and interferon induced T-cell alpha chemoattractant (I-TAC) (MCP-1; **p = 0.001 for infected and **p = 0.005 for infused at d2 relative to d0 using a one-tailed paired t test, IP-10; *p = 0.03 for infected and ***p = 0.0008 for infused at d2 relative to d0 using a one-tailed paired t test, ITAC; ***p = 0.0005 for infected and ***p = 0.0007 for infused at d2 relative to d0 using a one-tailed paired t test). **D** Correlation of innate immune cells against chemokines, and interleukin (IL)-10 vs IL-6 (two-tailed Pearson test p values shown, 95% confidence bands of the best fit line are shown).

It should be noted, however, that responses to S, N, M were also detected in splenocytes from opportunistic necropsies of SARS-CoV-2 unexposed animals, suggestive of cross-reactive T cells to endemic coronaviruses, as has been reported in humans[17,28]. Indeed, natural infection of colony-bred rhesus macaques with HCoV-NL63 has been documented[29]. With respect to vaccine outcomes, whether these cross-reactive T cells are able to recognize and mount effector responses against SARS-CoV-2-infected cells in vivo is highly relevant, as pre-existing cross-reactive T cells could impact vaccine efficacy.

Evaluation of CD4 T-cell polyfunctionality in the spleen by ICS in response to PMA/Ionomycin stimulation revealed that CXCR5+ CD4 T cells were clearly distinguishable from CXCR5− subsets in their ability to co-produce IFNγ, IL-2, TNFa, and IL-21. In contrast, the CXCR5− subset produced little IL-21 yet was able to co-produce IL-2 and TNFa, or, alternatively, either IFNγ, IL-2, or TNFa. In contrast, CD8 T cells were predominantly IFNγ producers (Fig. 4C, D). T cells from uninfected animals showed a similar representation of polyfunctionality indicating that SARS-CoV-2 infection did not alter polyfunctional profile of T cells to a mitogenic stimulus. We also identified modest frequencies of IFNγ and TNFa producing SARS-CoV-2-specific CD4 $T_{fh}$ cells in the spleen (Figure S3C, D) Consistent with splenic data, antigen-specific responses against S and N were also observed in peripheral blood at Day 7 (Fig. 5A, B). Together, these data demonstrate that S- and N-specific CD4 $T_{fh}$ cells are elicited following SARS-CoV-2 infection.

**SARS-CoV-2 infection induces GC responses in mediastinal lymph nodes.** Having established that SARS-CoV-2 stimulates the production of CD4 $T_{fh}$ cells, we next assessed whether $T_h1$ effector CD4 T cells were induced in the lung. Subsequent to collagenase digestion, single-cell suspensions isolated from the lung were stained with a panel of markers to delineate activated CD4 T cells. We evaluated the expression of Granzyme B and PD-1, both antigen-induced activation markers; mucosal homing receptors $a_4β_7$, CCR6, and the $T_h1$ receptor CXCR3 within CD69+ and CD69− CD4 and CD8 T-cell subsets (Fig. 6A). The expression pattern of Granzyme B, PD-1, and CXCR3 in lung CD4 T cells was indicative of a $T_h1$ effector CD4 response consistent with evidence for CD8 T-cell effectors in the lung parenchyma (Fig. 6B). We also observed that the proportion of Granzyme B+ and PD-1+ CD8 T cells correlated with viral loads in the nasal tract, indicative of antigen-driven expansion of CD8 T cells within the lung (Fig. 6C). Furthermore, histopathology of the lung showed the development of Bronchus-associated lymphoid tissue (Figure S4A) substantiating induction of adaptive immune responses. Gross examination of the mediastinal lymph nodes was consistent with lymphadenopathy (Figure S4B). Correspondingly, H&E-stained mediastinal lymph node sections showed distinct abundance of pale GCs (Figure S4C). Immunostaining of formalin-fixed lymph node sections showed CD3+ PD-1-expressing cells (Fig. 7A, e, i) localized to lymphoid follicles consisting of CD20+ Bcl-6+ cells (Fig. 7A, e, d) confirming the presence of GCs in the mediastinal lymph node following SARS-

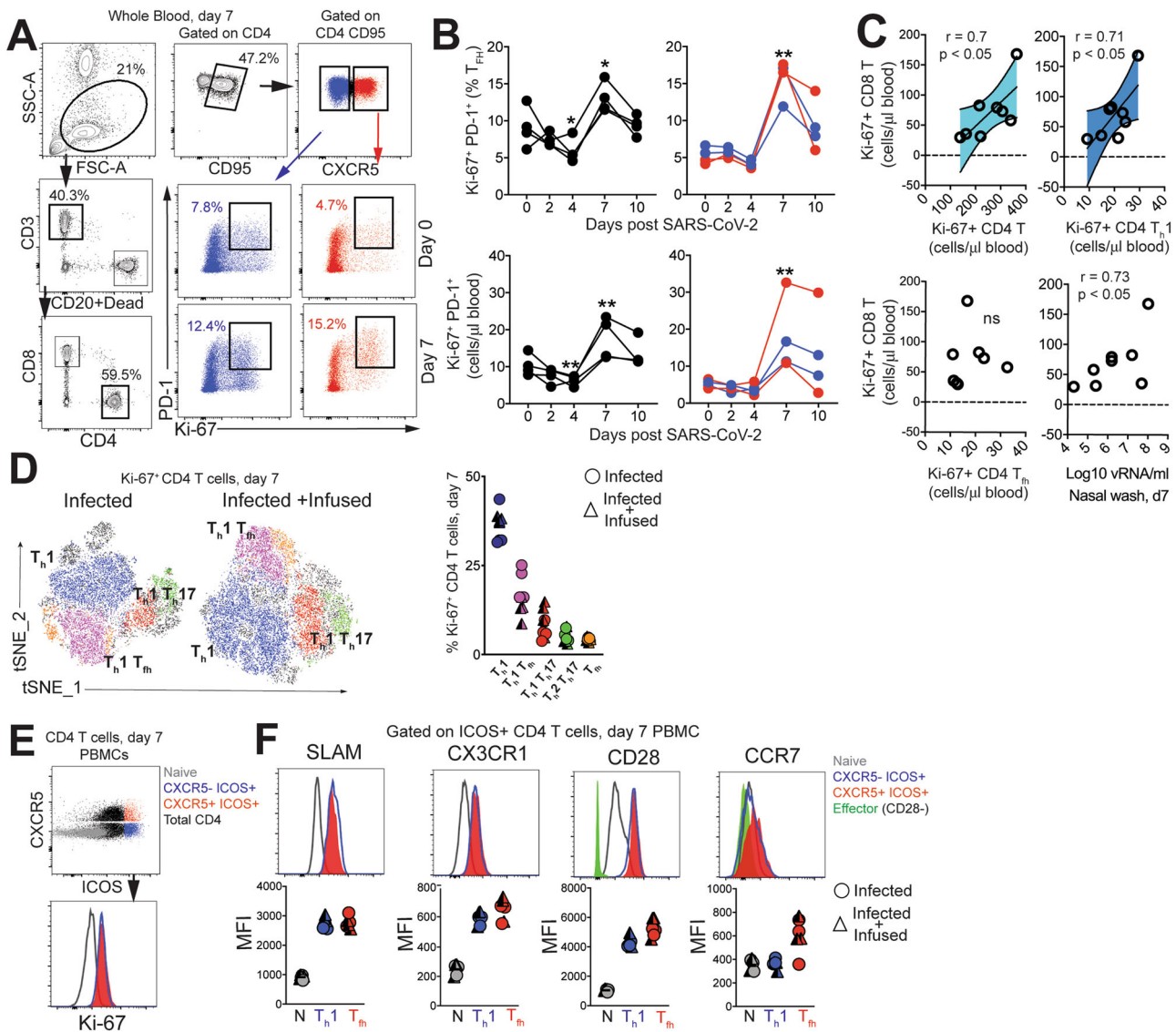

**Fig. 3 SARS-CoV-2 infection increases the number CD4 T follicular helper cells in peripheral blood. A** Representative gating strategy to capture CD4 T cells expressing Ki-67 and programmed death-1 (PD-1) in whole blood. Fluorochromes used were CD3-A700, CD20/Dead-APC-Cy7, CD8-BUV 805, CD4-BV650, CD95-BUV737, CXCR5-PE, PD-1-Pe Cy7, Ki-67-A488, CXCR3-BV786, CCR6-PECF594, CCR4-BV605, SLAM-A488, CX3CR1-PECF594, CD28-Pe-Cy7, CCR7-BV711, ICOS-BV786. **B** Kinetics show frequency and absolute counts of Ki-67$^+$ PD-1$^+$ CD4 T follicular helper cells (T$_{fh}$) cells (% of T$_{fh}$ cells; *$p = 0.01$ at d4 and d7 relative to d0 for infected and **$p = 0.002$ at d7 relative to d0 for infused using a one-tailed paired $t$ test, absolute T$_{fh}$ cell counts; **$p = 0.003$ at d4 and **$p = 0.0086$ at d7 relative to d0 for infected and **$p = 0.003$ at d7 relative to d0 for infused using a one-tailed paired $t$ test. Data are from a real-time longitudinal staining of whole blood performed a single time) **C** correlation plots of Ki-67$^+$CD8 T cells against Ki-67$^+$ CD4 subsets, and viral(v)RNA (all day 7) (two-tailed Pearson test $p$ values shown. 95% confidence bands of the best fit line are shown) **D** t-distributed stochastic neighbor embedding (tSNE) plot based on flow cytometry data of CD4 Ki-67$^+$ events at Day 7 from infected (16,197 events) and infected + infused animals (22,406 events); dot plot shows frequency of Ki-67$^+$ CD4 T-cell subsets. (**E-F**) Histograms and median fluorescence intensity (MFI) dot plots illustrate relative expression of signaling lymphocyte activation molecule (SLAM), CX3C chemokine receptor 1 (CX3CR1), CD28, and C-C chemokine receptor type 7(CCR7) within four different populations identified at Day 7 in peripheral blood mononuclear cells (PBMCs, $n = 7$). Unique symbols identify animals in each of the experimental groups.

CoV-2 infection. To quantitatively assess GC responses, we obtained single-cell suspensions from lymph nodes and stained cells with a panel of markers to define GC T$_{fh}$ cells, GC B cells, and follicular dendritic cells (FDCs). As illustrated, mediastinal lymph nodes showed a distinct CXCR5$^+$ PD-1$^{++}$ GC T$_{fh}$ subset and Bcl-6$^+$ CD95$^+$ CD20$^+$ GC B cells (Fig. 7B). FDCs were identified based on expression of the complement receptor CD21 (clone B-Ly4; Figure S5A), within the CD45$^-$ CD3$^-$ CD20$^-$ cell population[30]. The number of FDCs strongly correlated with the frequencies of both GC B cells and GC T$_{fh}$ cells (Figure S5B).

Quantifying the expression of canonical GC markers showed that Bcl-6 was exclusively expressed by GC B cells and to a lesser extent by GC T$_{fh}$ cells (Fig. 7C). FDC markers CD21 and platelet-derived growth factor receptor b (CD140b)[31] were also expressed by GC T$_{fh}$ and B cells. An increase in expression of the T$_{h}$1-chemokine receptor, CXCR3, on GC T$_{fh}$ cells was consistent with the phenotype of cells responding to viral infection. Although GC B cells displayed heterogeneity in CXCR3 expression, FDCs were uniformly negative for this marker. The increased number of GC T$_{fh}$, GC B cells, and FDCs (Fig. 7D), as well as the higher relative

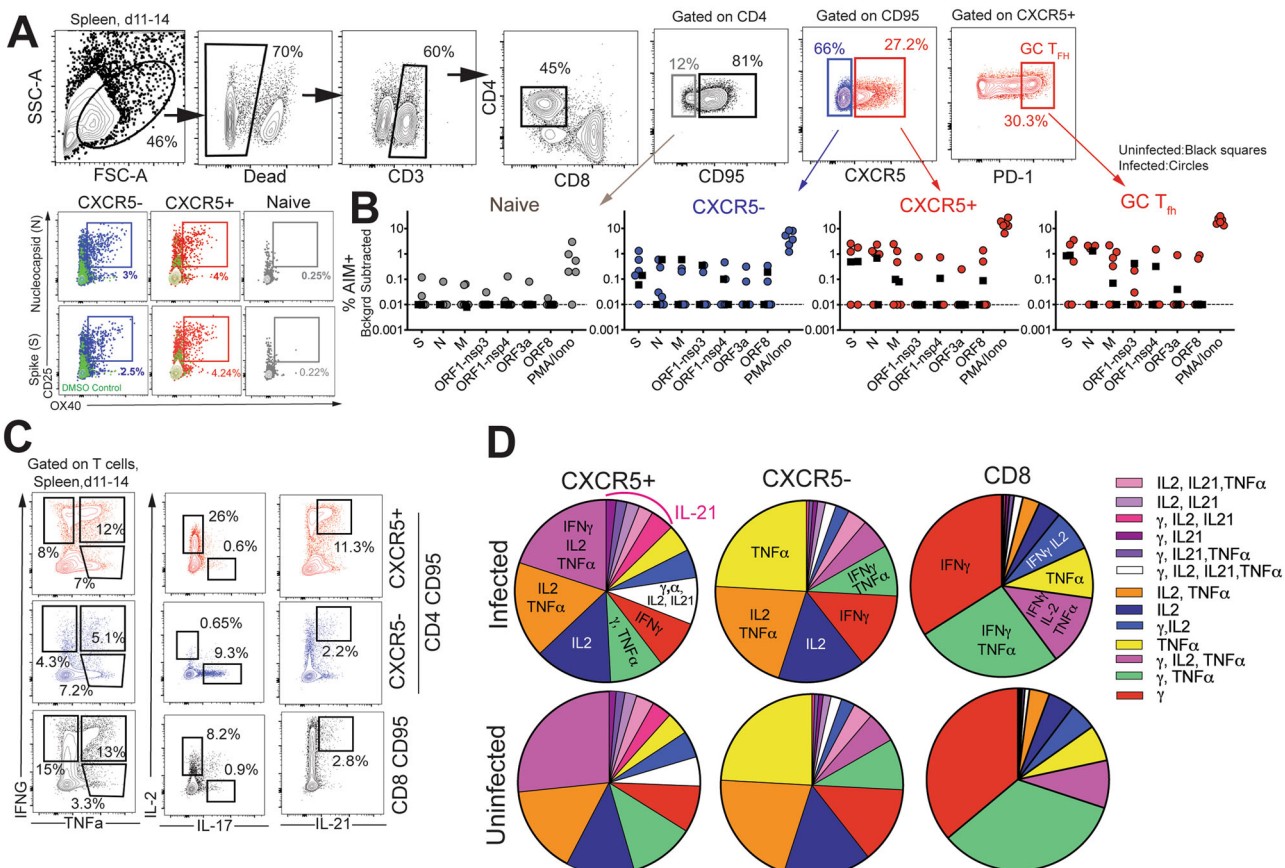

**Fig. 4 CD4 T$_{fh}$ cells targeting spike (S) and nucleocapsid (N) are generated in lymphoid tissue following SARS-CoV-2 infection. A** Representative gating strategy to identify SARS-CoV-2-specific CD4 T cells following stimulation with peptide megapools; membrane (M), open reading frame non-structural proteins (ORF-nsp) and Phorbol 12-myristate 13-acetate (PMA)/Ionomycin (Iono) Fluorochromes used were CD3-A700, Dead-APC-Cy7, CD8-BV510, CD4-BV650, CD95-BUV737, CXCR5-PE, PD-1-Pe Cy7, CD25-APC, OX40-BV786, IFNG-Pe-Cy7, TNFa-A488, IL-17-BV421, IL-21-APC. **B** Scatter plot showing Activation-induced marker (AIM) + CD4 subsets. Dashed line represents undetectable responses assigned a value of 0.01% **C** Gating strategy to identify cytokine profiles (interferon (IFN)γ, interleukin (IL)-2, tumor necrosis factor (TNF)a, interleukin (IL)-17, interleukin (IL)-21) of CXCR5$^+$, CXCR5-, and CD8$^+$CD95$^+$ T cells) in spleen following stimulation. **D** Pie chart shows T-cell polyfunctionality.

expression of CXCR3 in mediastinal lymph nodes compared to cervical and mesenteric lymph nodes, indicated an active immune response to viral infection (Fig. 7E, S5C). Functional analysis of GC T$_{fh}$ cells in the mediastinal lymph node following PMA/Ionomycin stimulation showed enrichment of CD40L$^+$ IFNγ $^+$ cells within the GC T$_{fh}$ compartment, consistent with a T$_h$1 phenotype within the mediastinal lymph node (Figure S5D).

To understand how CXCR3 expression within GC T$_{fh}$ cells might influence localization and helper potential, we performed phenotypic characterization of CXCR3$^-$ and CXCR3$^+$ GC T$_{fh}$ cells following SARS-CoV-2 infection (Fig. 8A–C). The data showed equivalent expression of CXCR5 and CCR7, key receptors that control follicular localization among both CXCR3$^-$ and CXCR3$^+$ subsets[32]. Furthermore, both subsets expressed comparable levels of CD69 on a per-cell basis. Consistent with observations in pT$_{fh}$ cells, SLAM expression was higher in the CXCR3$^+$ GC T$_{fh}$ subset[13]. Expression of ICOS was comparable while CXCR3$^+$ GC T$_{fh}$ cells expressed higher levels of CD28. Functional analysis following PMA/Ionomycin stimulation revealed that CXCR3$^+$ GC T$_{fh}$ cells expressed higher levels of IFNγ and were comparable for expression of CD40L and IL-21 (Fig. 8C). Akin to responses in spleen and PBMCs, SARS-specific responses by GC T$_{fh}$ cells in the mediastinal lymph node were observed (Fig. 8D). Together, these data indicate that SARS-CoV-2 infection induces GC T$_{fh}$ cells in otherwise healthy rhesus macaques.

**Humoral responses to SARS-CoV-2 are dominated by IgG antibodies**. The kinetics of the early antibody response to S and N proteins and the contributing antibody isotypes, specifically in the setting of mild or asymptomatic clinical illness, are not well-defined. Here, we quantified concentrations of serum antibodies to S1, S2, and N antigens, and used a secondary antibody specific for macaque IgG to distinguish de novo IgG antibodies from passively infused human CP IgG antibodies. The data showed IgM and IgG seroconversion to S1 and S2 proteins in all animals by day 7 post-infection, with the exception of one animal (Fig. 9A, B). This is consistent with reports that S- or RBD-specific IgG and IgM antibodies often appear simultaneously in blood of most humans infected with 2002 SARS or CoV-2[33–35]. Antibody responses to the N protein in humans are reported to increase 10 days following disease onset[36,37], and interestingly, N-specific IgG was evident in all macaques by day 7 but N-specific IgM was not increased significantly (threefold over baseline values) until day 10 in most animals. In addition, 50% of the animals failed to demonstrate a significant IgA response to all SARS-CoV-2 proteins within 10 days of infection (Fig. 9C). However, we should note that analysis of some necropsy sera suggested that IgA antibodies continue to increase after day 10 (Fig S5E).

Evaluation of the magnitude of post-infection antibody responses in animals that did not receive CP plasma clearly

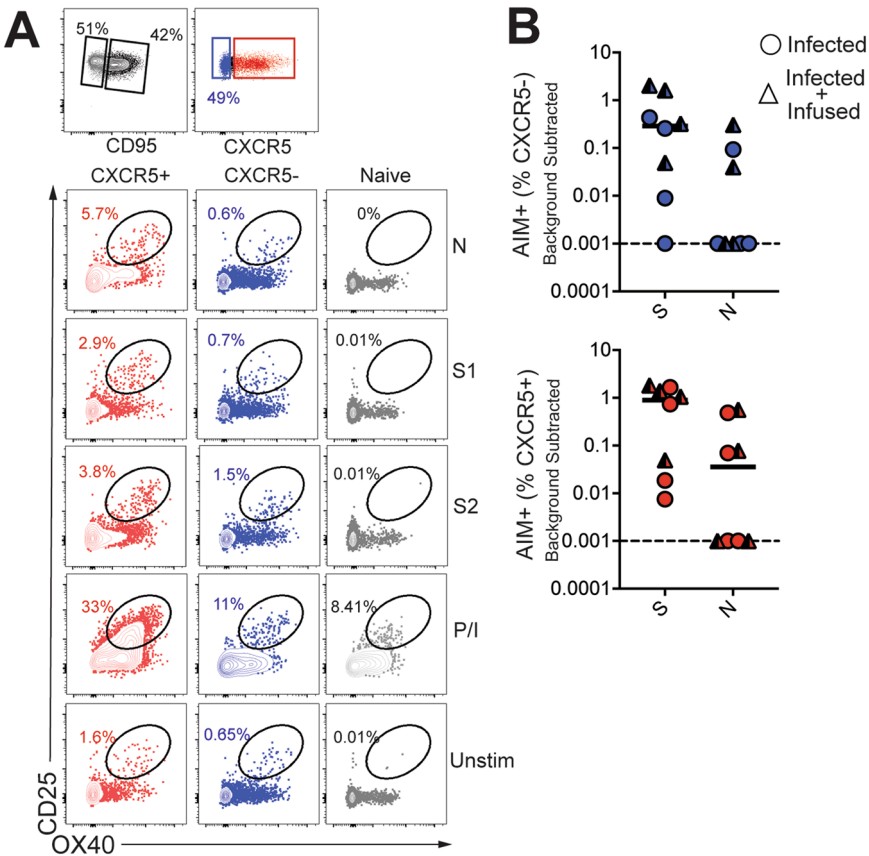

**Fig. 5 CD4 T$_{fh}$ cells targeting spike (S) and nucleocapsid (N) in blood following SARS-CoV-2 infection. A** Representative gating strategy to identify SARS-CoV-2-specific CD4 T cells following stimulation with spike (S) and nucleocapsid (N) peptide pools in PBMCs. **B** AIM + CXCR5$^-$ and CXCR5$^+$ CD4 subsets in PBMCs at Day 7.

indicated that IgG dominated the humoral response to all SARS-CoV-2 proteins (Fig. 10A), with IgG1 being the most dominant subtype (Fig. 10B). On day 10, we observed strong correlations between S1-specific IgG and IgM and between N-specific IgA and IgG (Fig. 10C). The pseudovirus nAbs ranged from 82 to 1754, higher than those reported in some studies[23], with two out of eight animals showing a > 1:1000 neutralizing activity in sera indicative of productive humoral immune responses. Consistent with reports in infected humans, we observed a strong correlation between neutralization antibody titers and concentrations of anti-RBD IgG antibodies on day 10 (Fig. 10C). Furthermore, the proportion of T$_h$1 T$_{fh}$ effector cells at day 7 was predictive of anti-RBD IgG concentrations at day 10 (Fig. 10C). Together, these data show the development of binding and neutralizing antibodies following SARS-CoV-2 infection in the context of mild or absent clinical symptoms. The appearance of antiviral IgG antibodies by day 7 with delayed induction of IgA responses suggests that early class switching occurs after SARS-CoV-2 infection and is likely promoted by T$_h$1-type T$_{fh}$ cells. However, it should be noted that mucosal IgA responses can be induced in a T-independent manner[38].

## Discussion
The importance of CD4 T$_{fh}$ cells in the generation of plasma cells, critical for persistent antibody, places a premium on understanding the T$_{fh}$ and GC response following SARS-CoV-2 infection. The present study adds to our understanding of immune responses to SARS-CoV-2 in three significant ways. First, we demonstrate that robust T$_h$1–T$_{fh}$ responses are observed following SARS-CoV-2 infection. Second, T$_{fh}$ responses focused on S

and N are seen within lymph nodes, circulate through peripheral blood, potentially seeding the spleen. At last, we show that acute antibody kinetics are characterized by induction of IgG, predominantly to S1 and of the IgG1 subclass, indicative of early class switching. Taken together, these data demonstrate that productive T$_{fh}$ responses are elicited following SARS-CoV-2 infection in healthy adult rhesus macaques. While the T$_h$1-bias of T$_{fh}$ cells following infection is expected due to the robust interferon response following SARS-CoV-2, this skewing is important to note, as weak interferon responses observed in COVID patients could hamper shoring up effective antiviral antibody and CD8 T-cell responses[39].

Several studies have examined the kinetics of antibody responses in humans after the onset of symptoms and three unifying themes emerge from these data. First, in the majority of patients, antibodies to RBD of the S1 subunit are induced between 8 and 10 days of symptom onset, and levels of these antibodies correlate strongly with neutralizing titers[34,35,40]. Second, plasma from the majority of COVID-19 convalescent patients do not contain high levels of neutralizing activity[41]. Third, plasma antibodies in infected individuals that do develop neutralizing antibodies are minimally mutated[42]. These data suggest that CD8 T cells may contribute to the control of SARS-CoV-2, and while a protracted GC response may not be critical for the generation of neutralizing antibodies it could improve antibody durability by enhancing plasma cell numbers. Our data add to this developing narrative by showing that in the setting of mild/asymptomatic illness, antibody responses are generated and characterized by the predominance of IgG. Intriguingly, we observed that IgM and IgG antibodies to the S1, S2, and N proteins were produced concurrently.

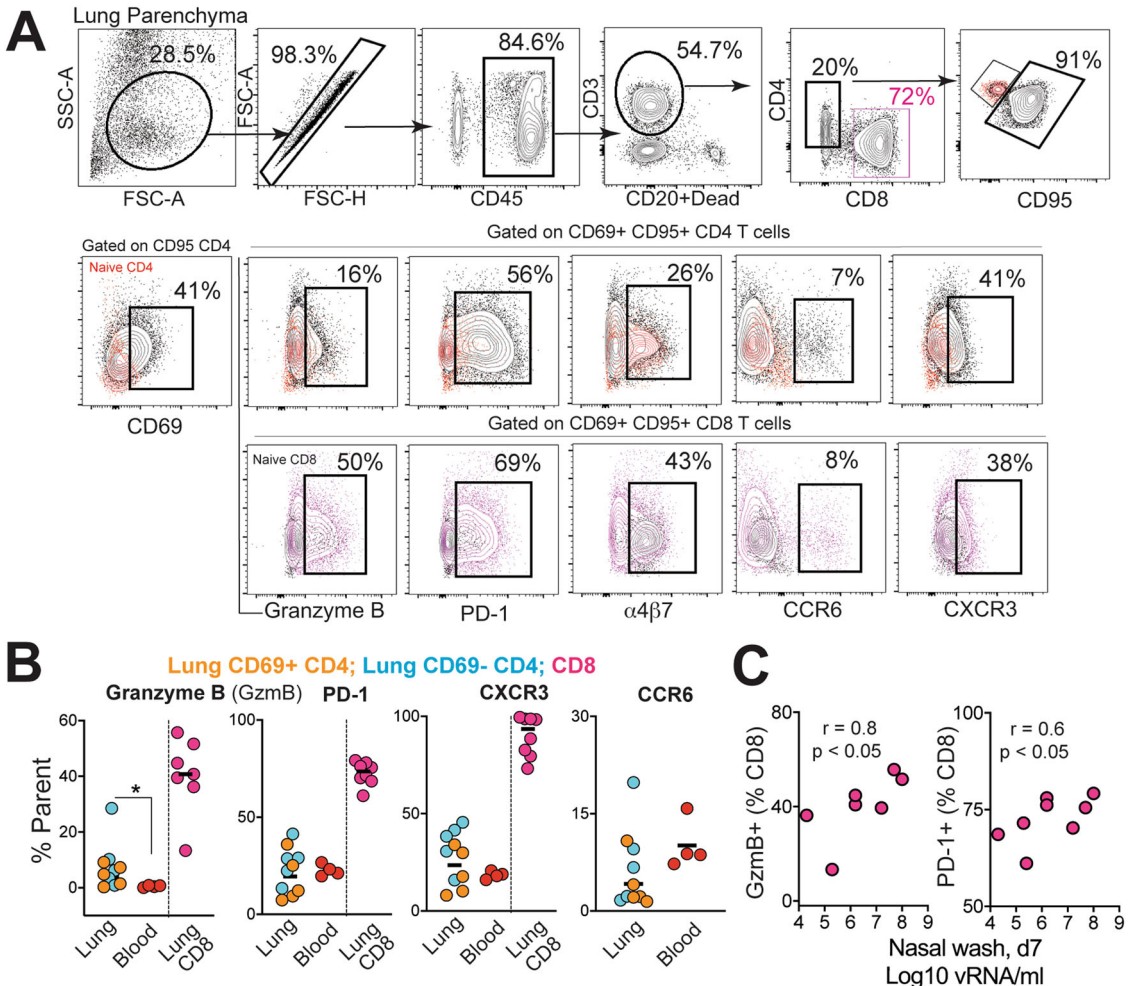

**Fig. 6 Induction of T$_h$1 CD4 effectors in the lungs during SARS-CoV-2 infection. A** Gating strategy for identifying CD95[+] CD69[+] CD4 and CD8 cells expressing granzyme B, PD-1, α4β7, CCR6, and CXCR3. Fluorochromes used were CD45-A488, CD3-A700, CD20/Dead-APC-Cy7, CD8-BUV 805, CD4-BV650, CD95-BUV737, CD69-BV711, Granzyme B- BV421, PD-1-Pe Cy7, a4b7-PE, CD25-APC, CCR6-PECF594, CXCR3-BV786. **B** Percentage of CD4 and CD8 T cells expressing granzyme B, PD-1, CXCR3, and CCR6 in lung and blood (*$p = 0.02$ using a two-tailed Mann–Whitney $U$ test). **C** Correlation plot of vRNA from nasal washes and either granzyme B (GzmB) or PD-1 in CD8 T cells (one-tailed Pearson test $p$ values shown).

Recent data suggest a benefit of CP therapy, in conjunction with antiviral and immunomodulators, in treating moderate to severe COVID-19[43,44] but also show that CP infusion early during the course of infection may be more beneficial as antibody responses are generated within 2 weeks of symptom onset[45]. Our studies, however, demonstrate that CP infusion did not abate a nascent infection. While the pooled CP had neutralization titers of 1:1149, we failed to achieve detectable neutralizing antibodies after infusion explaining the lack of efficacy. These results highlight the prerequisite for convalescent plasma to have sufficiently high titers of neutralizing antibodies to treat COVID-19 patients. More controlled studies with sufficient animal numbers are required to conclusively determine whether high-titer CP infusion, or multiple CP infusions, achieve neutralizing titers that can protect either prophylactically or therapeutically. Data supporting this would have considerable potential not only for therapeutics but also for efficacy of vaccines.

Although it remains unknown whether immune responses elicited when naturally infected by SARS-CoV-2 will protect from re-infection, studies in rhesus macaques show that infection does protect against re-challenge, 28–35 days post first infection, signifying that some degree of a protective immune response follows infection[23]. In this context, the finding that infection induces

CD4 helper responses targeting major structural proteins on the virus suggests that infection is capable of producing effective CD4 help for CD8 T cells and antibody responses. Indeed, antibodies against the RBD region in S1 are elicited in the vast majority of COVID-19 patients along with robust CD4 T-cell responses[35]. Our data show that spike epitopes are immunogenic to both T and B cells and suggest that induction of these responses by vaccines may confer protection. Although our discovery of N- and S-specific CD4 T cells in the spleen is intriguing; at present we are unable to distinguish whether these cells represent cells that are seeded from circulation or are elicited de novo via trafficked antigen. Further studies are needed to tease apart the possibilities as this is central to understanding the determinants of protective immunity. In sum, the data suggest that vaccine platforms inducing T$_h$1 CD4 helper and T$_{fh}$ helper responses are likely to succeed in eliciting robust CD8 T-cell and antibody responses against SARS-CoV-2. Similar observations of circulating T$_{fh}$ cells in vaccinated and infected humans support this hypothesis[16,46,47]. However, it must be emphasized that these are associations, and the necessity for T$_h$1 CD4 T cells in the antibody response cannot be addressed in the present study.

In summary, the current study adds to our understanding of the CD4 helper responses to SARS-CoV-2 infection and provides

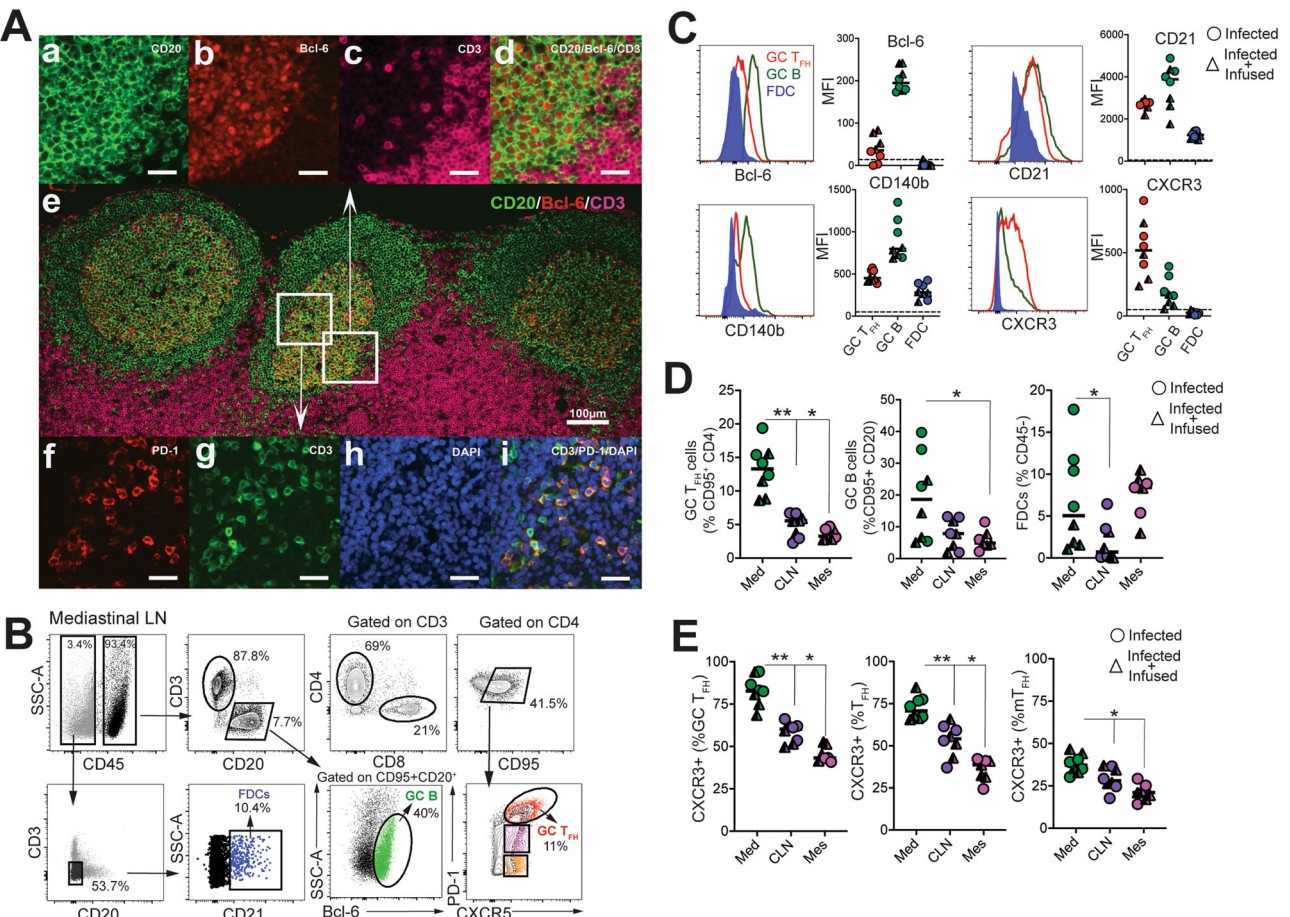

**Fig. 7 SARS-CoV-2 infection induces germinal center responses in mediastinal lymph nodes. A** Representative multi-color immunofluorescence image of CD3, PD-1, CD20, Bcl-6 with DAPI staining in mediastinal lymph nodes. Two connecting sections were stained with CD3/PD-1 and CD20/ Bcl-6/CD3 to visualize germinal center (GC) $T_{fh}$ cells and GC B cells, respectively. Images in (a–d) showing GC B cells and images in (f–i) showing GC $T_{fh}$ cells are enlarged from white boxes in (e) and collected using a ×20 objective. Merged image in (d) shows CD20+Bcl-6+ GC B cells and image in (i) shows CD3 + PD-1+ GC $T_{fh}$ cells. Scale bar in (e) is 100 μm and the rest are 25 μm. CD3 stain in pink is pseudo color (original red) to distinguish from Bcl-6. **B** Representative gating strategy to identify follicular dendritic cells (FDC), germinal center B cells (GC B), and germinal center $T_{fh}$ cells (GC $T_{fh}$) in the mediastinal lymph nodes (Med) Fluorochromes used were CD45-A488, CD3-A700, CD20-BV421, Dead-BV510, CD8-BUV 805, CD4-BV650, CD95-BUV737, CXCR5-PE, PD-1-Pe-Cy7, Bcl-6-APC-Cy7, CD140b-APC, CD21-PECF594, CXCR3-BV786. **C** Median fluorescence intensity of Bcl-6, CD21, CD140b, and CXCR3. **D** Frequency of GC $T_{fh}$ cells, GC B cells, FDCs significantly higher in mediastinal lymph node (Med, data shown from $n = 8$ independent animals (GC $T_{fh}$; **$p = 0.007$, *$p = 0.01$) relative to cervical lymph nodes (CLN, data shown from $n = 8$ independent animals) and mesenteric lymph nodes (Mes, data shown from $n = 7$ independent animals) using a two-tailed Wilcoxon matched-pairs signed-rank test, GC B cells; *$p = 0.04$ using a two-tailed Wilcoxon matched-pairs signed-rank test, FDCs; $p = 0.039$ using a two-tailed Wilcoxon matched-pairs signed-rank test. Horizontal line indicates median. **E** Majority of GC $T_{fh}$ cells in mediastinal lymph nodes express CXCR3 (GC $T_{fh}$ and $T_{fh}$; **$p = 0.007$, *$p = 0.01$, and mTfh *$p = 0.01$ relative to CLN and Mes using a two-tailed Wilcoxon matched-pairs signed-rank test). Data shown are from $n = 8$ independent animals for Med, CLN, and $n = 7$ independent animals for Mes. Horizontal line indicates median.

an important foundation for harnessing the mechanisms that stimulate robust CD4 $T_{fh}$ responses in the context of an effective vaccine.

## Methods

**Rhesus macaques.** Eight colony-bred Indian origin rhesus macaques (*Macaca mulatta*) were housed at the California National Primate Research Center and maintained in accordance with American Association for Accreditation of Laboratory Animal Care guidelines and Animal Welfare Act/Guide. The study received ethical approval by the Institutional Animal Care and Use Committee at UC Davis and research personnel and animal staff complied with all relevant ethical regulations for animal testing and research. As described by the authors[48], strict social distancing measures were implemented at the CNPRC at the start of the pandemic in March to reduce the risk of human-to-rhesus SARS-CoV-2 transmission. Animals were screened for SARS-CoV-2 and housed in barrier rooms with increased PPE requirements prior to study assignment. Animals were four to five years of age with a median weight of 8.6 kg (range 5.4–10.7 kg), were SIV- STLV- SRV-. Animals were seronegative for SARS-CoV-2 at study initiation.

Sex distribution within experimental groups was as follows; infected ($n = 3$ females, $n = 1$ male); infected + convalescent plasma ($n = 2$ males); infected + normal plasma ($n = 2$ males). Table S1 provides details of the animals in the study. For blood collection, animals were anesthetized with 10 mg/kg ketamine hydrochloride injected i.m. For virus inoculation and nasal secretion sample collection, animals were additionally anesthetized with 15−30 μg/kg dexmedetomidine HCl inject i.m. and anesthesia was reversed with 0.07–0.15 mg/kg atipamezole HCl injected i.m.

**Virus and inoculations.** SARS-CoV-2 virus was isolated from the nasal swab of a COVID-19 patient with acute respiratory distress syndrome admitted to the University of California, Davis Medical Center, Sacramento[49]. The nasal swabs were obtained from the UC Davis Biorepository as approved by the local Institutional Review Board. The clinical samples were de-identified by the health system laboratory prior to release and complied with relevant ethical and human subjects research regulations. Vero cells (ATCC CCL-81) were used for viral isolation and stock expansion. The passage 2 viral stock (SARS-CoV-2/human/USA/CA-CZB-59×002/2020) used for animal inoculations had a titer of $1.2 × 10^6$ PFU/ml (corresponding to $2 × 10^9$ vRNA) (Genbank accession number: MT394528). To recapitulate relevant transmission routes of SARS-CoV-2, animals were inoculated with

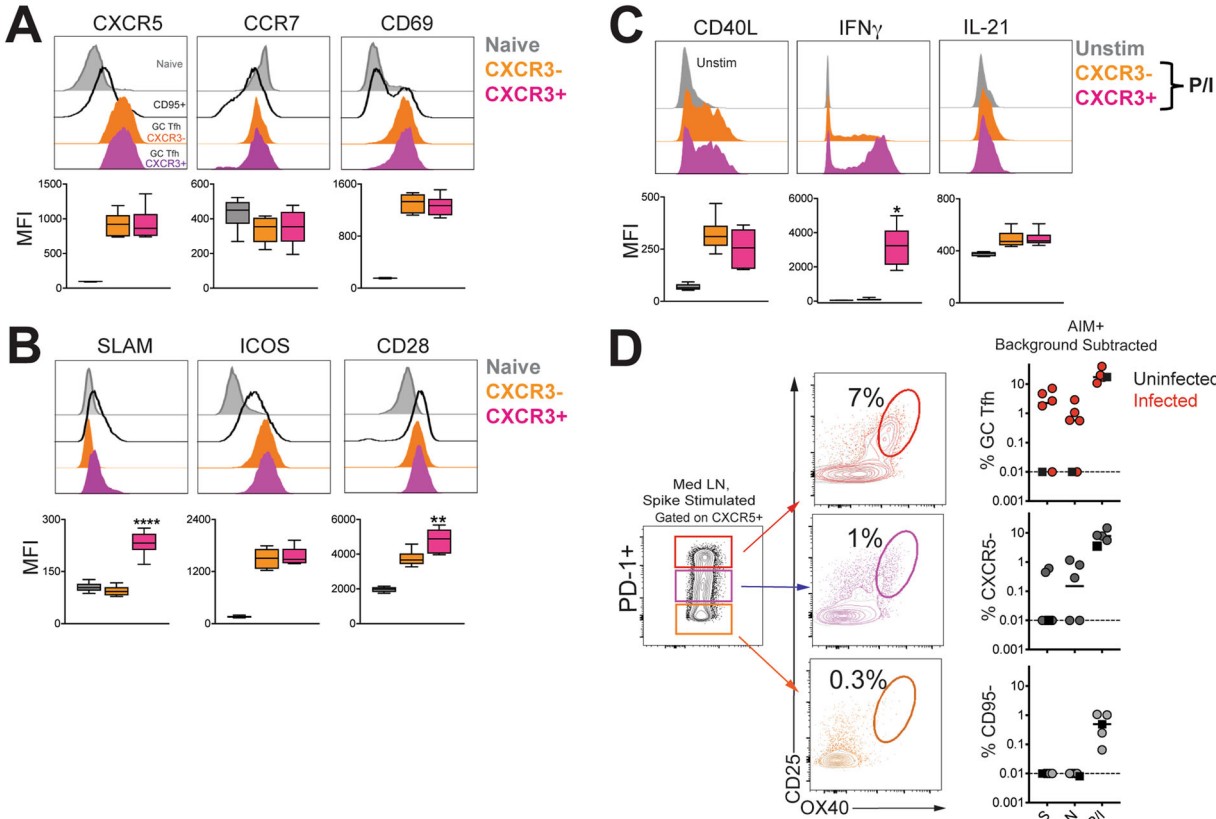

**Fig. 8 SARS-CoV-2 infection induces germinal center responses targeting spike (S) and nucleocapsid (N) in mediastinal lymph nodes. A** Median fluorescence intensity (MFI) of CXCR5, CCR7, CD69 and **B** SLAM, ICOS, CD28 within CXCR3- (orange) and CXCR3+ (magenta) GC T$_{fh}$ cells in spleen following SARS-CoV-2 infection. Naive CD4 T cells in spleen shown for comparison (grey) (SLAM; ****$p$ < 0.0001 CXCR3+ relative to naive and CXCR3− using a paired two-tailed $t$ test, CD28; **$p$ = 0.001 CXCR3+ subset relative to naive and CXCR3- using a paired two-tailed $t$ test) **C** Following PMA/ Ionomycin stimulation, CD40L, IFNγ, and IL-21 expression shown across GC T$_{fh}$ subsets. Unstimulated cells shown in gray (IFNG; **$p$ = 0.0012 CXCR3+ relative to CXCR3− using a paired two-tailed $t$ test). Box-whiskers plot shows range of data, bounds of the box extend from the 25th to 75th percentile, line in box is plotted at the median. **D** Flow plot of PD-1+ CXCR5+ T$_{fh}$ subsets shows AIM+ cells following stimulation with spike (S); scatter plot shows specificity of GC T$_{fh}$ cells and CXCR5− cells to SARS-CoV-2 S and nucleocapsid (N), and responses to PMA/Ionomycin. CD95- naive CD4 T cell shown for comparison. The dashed line represents undetectable responses assigned a value of 0.01%. Black squares denote SARS-CoV-2 unexposed animals. Fluorochromes used were CD40L-APC; remaining as stated in previous panels.

1 ml stock instilled into the trachea, 1 ml dripped intranasally, and a drop of virus stock in each conjunctiva.

**Convalescent plasma and infusions**. Convalescent plasma was sourced from Vitalant and represented a pool of up to four donors. Plasma was pooled prior to infusion into monkeys. Pooled plasma had a nAb titer of 1:1149, binding antibody titers for SARS-CoV-2 antigen were as follows; anti-S1-IgG, 24.5 µg/ml; anti-S2 IgG, 2.9 µg/ml; anti-N IgG, 10.7 µg/ml. Normal plasma was collected prior to the COVID-19 pandemic and was negative for SARS-CoV-2 antibody. Concentrations were as follows; anti-S1-IgG, 0.00004 µg/ml; anti-S2 IgG, 0.003 µg/ml; anti-N IgG, 0.001 µg/ml. Twenty fours following virus inoculation, animals were infused with plasma at 4 ml/kg volume (total volume infused was 33−39 ml) at an infusion rate of 1 ml/minute. Control animals ($n = 4$) were not infused.

**Specimen collection and processing**. On days 2, 4, 5, 7, 8, and 10, a five-french feeding tube was inserted into the nasal cavity. In all, 2 ml PBS was instilled through each nostril and the maximum volume was aspirated, secretions were spun at 800 × $g$ for 10 min and 1 ml of supernatant and cell pellet were lysed in 3 ml Trizol LS for RNA isolation. Ethylenediaminetetraacetic acid-anticoagulated blood was collected on Days 0, 2, 4, 7, and 10 for immunophenotyping. PBMCs were isolated from whole blood collected in cell preparation vacutainer tubes, sampled at Day 7 and necropsy, by density gradient centrifugation[50]. For serum, coagulated blood was centrifuged at 800 g for 10 min to pellet clotted cells, followed by extraction of supernatant and storage at −80 °C. Lymph nodes, spleen, and lung tissue were obtained at necropsy and digested enzymatically using collagenase followed by manual disruption to obtain single-cell suspensions for flow cytometry-based assays. Data from whole blood samples were generated in real-time and were repeated once.

**AIM assay**. Cells were stimulated with overlapping peptide pools representing SARS-CoV-2 and responding cells were identified by upregulation of activation markers, as described previously[50,51]. All antigens were used at a final concentration of 2 µg/mL in a stimulation cocktail made with using 0.2 µg of CD28 and 0.2 µg CD49d costimulatory antibodies per test. Unstimulated controls were treated with volume-controlled DMSO (Sigma-Aldrich). Tubes were incubated in 5% CO$_2$ at 37 °C overnight. Following an 18 h stimulation, the cells were stained, fixed, and acquired the same day. AIM assays on splenocytes and mediastinal lymph nodes were performed on cryopreserved cells (Table S2). AIM assay on day 7 PBMCs were performed on fresh cells. Phenotype panel on LNs and PBMCs was performed using standard flow cytometry assays.

**vRNA quantitation by qRT-PCR**. Trizol lysed nasal samples were processed using a modified Qiagen RNeasy Lipid Tissue Mini Kit protocol. In brief, 5.6 µl polyacryl carrier was added to trizol lysate, followed by 1/10 volume BCP and phase separated as described in Qiagen protocol. In all, 8 µl of eluted RNA was DNase treated with ezDNase per kit instructions and converted to cDNA with Superscript IV using random primers in a 20ul reaction and quantified in quadruplicate by qPCR on an Applied Biosystems QuantStudio 12 K Flex Real-Time PCR System using Qiagen QuantiTect Probe PCR Mastermix with primers and probe that target the SARS-CoV-2 Nucleocapsid (forward 5′-GTTTGGTGGACCCTCAGATT-3′, reverse 5′-GGTGAACCAAGACGCAGTAT-3′, probe 5′-/5-FAM/TAACCAGAA/ ZEN/TGGAGAACGCAGTGGG/3IABkFQ/-3′).

**Serum cytokines**. Luminex® (NHP Cytokine Luminex Performance Pre-mixed kit, R&D, FCSTM21) was performed to evaluate cytokines in rhesus macaque sera. The assay was performed according to the manufacturer's protocol. The beads for each sample, control, and standard curve point were interrogated in a Luminex® 200 dual laser instrument (Luminex, Austin, TX), which monitors the spectral

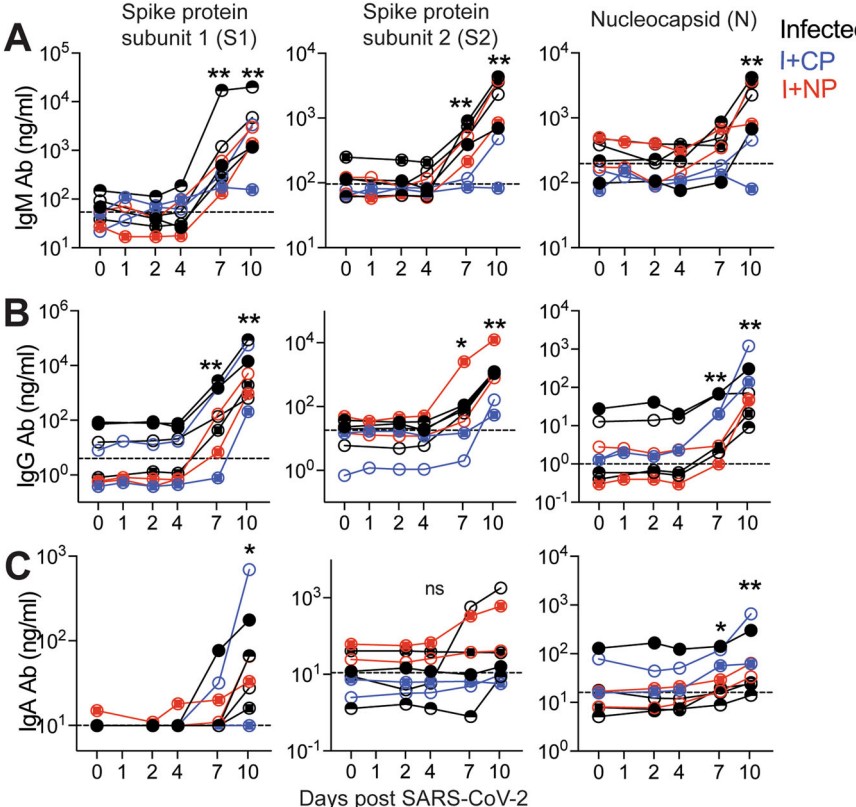

**Fig. 9 Humoral responses to SARS-CoV-2 are dominated by IgG antibodies.** Concentrations of **A** IgM, **B** IgG, and **C** IgA antibodies (Ab) specific for S1, S2, and N proteins measured by BAMA or ELISA in serum. The dashed line represents the median pre-infection (day 0) concentration for all animals. Unique symbols identify animals in each of the experimental groups. (**p = 0.007, *p = 0.015 at indicated time points relative to d0 using a Wilcoxon matched-pairs signed-rank two-tailed t test).

properties of the beads and amount of associated phycoerythrin (PE) fluorescence for each sample. xPONENT® software was used to calculate the median fluorescent index and calculate the concentration for each cytokine in

**Flow cytometry and immunofluorescent staining**. Whole blood and single-cell suspensions from the lung and lymph nodes were stained fresh and acquired the same day. Staining on spleen was performed on cryopreserved samples. Fluorescence was measured using a BD Biosciences FACSymphony™ with FACSDiva™ version 8.0.1 software. Compensation, gating, and analysis were performed using FlowJo (Version 10). For immunofluorescence imaging, 4um cut sections on slides were incubated with antibodies and goat anti-rabbit IgG conjugated to Alexa Fluor 568, goat anti-rat IgG conjugated with Alexa Fluor 488 (Molecular Probes, Grand Island, NY) was used to detect PD-1 positive cells and CD3⁺ T lymphocytes, respectively. Goat anti-mouse IgG2a conjugated to Alexa Fluor 488, goat anti-mouse IgG2b conjugated with Alexa Fluor 568, and goat anti-rabbit IgG conjugated with Alexa Fluor 647 (Molecular Probes, Grand Island, NY) were used to detect CD20⁺ B cells, Bcl-6⁺ cells, and CD3⁺ T lymphocytes, respectively. Coverslips were placed on all slides using the ProLong Gold reagent with 4′,6-diami-dino-2-phenylindole as a nuclear stain (Molecular Probes, Grand Island, NY). The slides were viewed, and images were captured with epifluorescent illumination using a Zeiss Imager Z1 microscope (Carl Zeiss Inc., Thornwood, NY) with an appropriate filter. Reagents used for flow cytometry and immunofluorescence are listed in Table S2.

Binding antibody multiplex assay (BAMA) for IgG and IgM antibodies to S1, S2 and N proteins A customized BAMA was developed to simultaneously measure antibodies to the following recombinant SARS-CoV-2 proteins (all from SinoBiologicals, Wayne, PA): S1 (#40591-V08H), S2 extracellular domain (#40590-V08B) and nucleocapsid (N; #40588-V08B). Briefly, proteins were dialyzed in PBS and conjugated to Bioplex Pro carboxylated magnetic beads (BioRad, Hercules, CA)[50]. Standard and serum samples treated with 1% TritonX-100 detergent were serially diluted in PBS containing 1% BSA, 0.05% azide, and 0.05% Tween-20 and mixed with beads overnight at 1100 rpm and 4 ˚C. The following day, the beads were washed and treated with biotinylated antibody followed by neutralite avidin-PE (Southern Biotechnology Associates: SBA, Birmingham, AL) as described (Phillips 2017). A BioRad Bioplex 200 and BioManager software were used to

measure fluorescent intensity and construct standard curves for interpolation of antibody concentrations in test samples.

The standard was pooled serum from macaques infected for 11–14 days with SARS-CoV-2. The following humanized (IgG1) monoclonal antibodies were used to estimate concentrations of IgG, IgM, and IgA antibodies in the rhesus serum standard: anti-S1 RBD (Genscript #HC2001), anti-S2 (SinoBiologicals #40590-D001) and anti-NC (Genscript #HC2003). Human and rhesus IgG antibodies were both detected in these calibration assays using biotinylated affinity-purified goat anti-human IgG γ chain polyclonal antibody (SBA #2048-08). In subsequent BAMA assays for SARS-CoV-2-specific rhesus macaque IgG antibodies, biotinylated mouse anti-monkey IgG γ chain monoclonal antibody (SBA cat#4700-08) was used as the secondary antibody. IgM antibodies were detected using biotinylated affinity-purified goat anti-human IgM μ chain polyclonal antibody (SBA#2020-08) which cross-reacts well with macaque IgM. Results obtained for IgM in the rhesus standard were multiplied by 3.3 to account for under-estimation by the monomeric IgG monoclonal antibody standard. Macaque IgA antibodies were detected with the antibodies described below. IgG, IgM, or IgA antibodies had to be increased threefold over the day 0 value to be considered significant.

For assays of human IgG antibodies in animals given convalescent plasma, serum from a COVID-19 convalescent human was used as standard. For assays of rhesus IgG and IgM antibodies, the standard was pooled serum from macaques infected with SARS-CoV-2. The following humanized (IgG1) monoclonal antibodies were used to estimate concentrations of antibodies in these standards: anti-S1 RBD (Genscript #HC2001), anti-S2 (SinoBiologicals #40590-D001) and anti-NC (Genscript #HC2003). Results for IgM were multiplied by 3.3 to account for under-estimation by the monomeric IgG antibody standard. Human or rhesus IgG antibodies in serum of animals were detected using NHP-adsorbed biotinylated affinity-purified goat anti-human IgG γ chain polyclonal antibody (SBA #2014-08) or biotinylated mouse anti-monkey IgG γ chain monoclonal antibody (SBA #4700-08), respectively. Both human and rhesus IgM antibodies were detected using biotinylated affinity-purified goat anti-human IgM μ chain polyclonal antibody (SBA #2020-08). The following rhesus IgG subclass-specific monoclonals (from the NHP Resource Reagent) were used to measure IgG1, IgG2, IgG3, and IgG4 antibodies: clone 3C10.3, 3C10, 2G11, and 7A8, respectively. These unlabeled monoclonal antibodies were detected using human-adsorbed biotinylated goat anti-mouse IgG2a or -IgG1 (SBA #1080-08 and #1070-08).

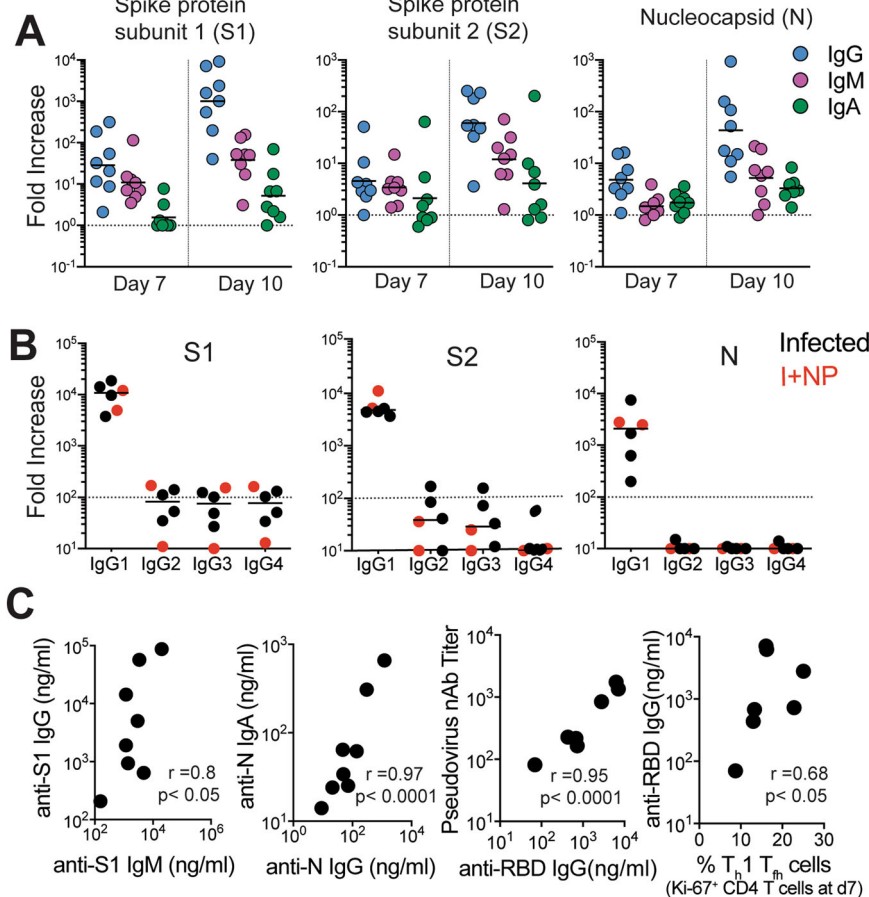

**Fig. 10 IgG1 subclass and neutralizing antibodies induced following SARS-CoV-2 infection. A** Fold increase in antibody responses in animals was determined by dividing post-infection concentrations by those measured on day 0 in each animal. Data shown are $n = 8$ animals for all time points. Horizontal line indicates median **B** Fold increase in IgG1, IgG2, IgG3, and IgG4 antibodies against S1, S2, and N show dominance of IgG1 subclass antibodies. Data shown are for $n = 6$ animals not given CP. **C** Correlations between day 10 levels of S1-specific IgG and IgM, N-specific IgA and IgG, and pseudovirus neutralizing antibody titers and anti-receptor binding domain (RBD) IgG antibodies measured by ELISA. Unique symbols identify animals in each of the experimental groups (two-tailed Pearson test $p$ values shown; correlation for anti-RBD IgG and $T_h1$ $T_{fh}$ cells shows one-tailed Spearman test $p$ value).

**Enzyme-linked immunosorbent assay for SARS-specific IgA and antibodies to RBD**. These assays were done using methods similar to those described[52] and Immulon 4 microtiter plates (VWR, Radnor, PA) coated with 100 ng per well of S1, S2, N, or RBD protein (SinoBiological #40592-VNAH). For IgA assays, a pooled rhesus serum collected day 14 after infection with SARS-CoV-2 was used as standard after depletion of IgG using GE Healthcare Protein G Sepharose (Sigma, St. Louis, MO) as described[52]. Test sera were also depleted of IgG to facilitate detection of low levels of IgA antibodies. Macaque IgA was detected using a mixture of biotinylated clone 10F12 (NHP Reagent Resource) and biotinylated clone 40.15.5 (Ward et al 1995) anti-rhesus IgA monoclonal antibodies, which do not cross-react with human IgA and, when combined, appear to recognize all allotypes of rhesus macaque IgA (Kozlowski, personal observation). For RBD IgG assays, the above-pooled rhesus serum standard and a secondary monoclonal antibody specific for macaque IgG were used. Neutralizing assay pseudovirus neutralization assay was performed as described[53].

**Statistics**. Statistical analyses were performed using GraphPad Prism 8.4.2. Within-group comparisons, such as immune responses and antibody levels at different time points, were done using the two-tailed Wilcoxon matched-pairs signed-rank test. For correlation analysis, the two-tailed Spearman rank correlation test was used. Graphical illustrations were made with Biorender.com

**Materials availability**. The study generated SARS-CoV-2/human/USA/CA-CZB-59×002/2020 and is available upon request from C.J.M. (https://www.ncbi.nlm.nih.gov/nuccore/MT394528 "MT394528").

**Reporting summary**. Further information on research design is available in the Nature Research Reporting Summary linked to this article.

## Data availability

The data sets generated and/or analysed during the current study are available in the figshare repository https://doi.org/10.6084/m9.figshare.13270325.

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

## Acknowledgements

We are grateful to Lourdes Adamson and Nicole Drazenovich for BSL-3 training and facilitating access to CIID BSL-3. The authors are grateful to Greg Hodges for facilitating the animal experiments in CNPRC ABSL3. We are extremely grateful to the primate center staff Wilhelm Von Morgenland, Miles Christensen, David Bennet, Vanessa Bakula, James Schulte, Jose Montoya, Joshua Holbrook, John McAnelly, Christopher Nelson, and David Bennett for animal sampling in ABSL3 at the CNPRC. Mark Allen provided necropsy technical expertise in the BSL-3 necropsy suite. The authors are grateful to Amanda Carpenter, Peter Nham, and Bryson Halley at the Primate Assay Laboratory Core. We thank Larry Dumont at VIR for the CP and Linda Fritts for assistance with PCR assays. We are grateful to David Asmuth for facilitating IRB approval to procure clinical samples. We thank Jeffrey Roberts, Lisa Miller, Marcelo Kuroda for their support in facilitating these experiments and acknowledge assay resources provided by the Primate Assay Laboratory Core and CNPRC base grant. The 10F12 anti-monkey IgA monoclonal antibody was obtained from the NIH Nonhuman Primate Reagent Resource supported by AI126683 and OD010976. This work was supported by internal seed grants to CNRPC and CIID, R21 AI143454-02S1 (S.S.I.), FAST GRANT—George Mason University (S.S.I.), NIH Contract # 75N9301900065 (A.S. and D.W.), and the CNPRC base grant P51OD011107.

## Author contributions

D.H.O.C., K.K.V.R., C.J.M., S.S.I. supervised the project. T.D.C., S.J.M., J.H.M., D.H.O.C., K.V.R., C.J.M., S.S.I. conceptualized study design. S.R.E., Y.S.L., J.W.R., B.A.S., A.V., N.K. N., S.S.I. performed flow assays. S.S.I. analyzed and interpreted flow data and S.R.E., Y.S. L., J.W.R. checked gates. R.L.S. performed clinical assessments. T.D.C., J.D.D. grew up virus stock and processed nasal washes. J.D. performed vRNA analyses using qRT-PCR. J.R.R., K.L.O., Z.M.M. performed histopathological analysis. Z.M.M. performed immunofluorescence staining and analysis. J.W., J.U., R.I., J.L.Y. conducted assays and assisted with necropsy. D.W., A.S., M.S., N.K.T., G.S., M.P.B. provided key reagents. P.A.K., K.D. W., S.F., and J.C.S. performed antibody assays and S.F. performed the pseudovirus neutralization assays. S.S.I. wrote the original draft of the manuscript and the revision. All authors reviewed and edited the manuscript.

## Competing interests

A.S. is listed as inventors on a provisional patent application covering findings reported in this manuscript. A.S. is a consultant for Gritstone, Flow Pharma, and Avalia. The remaining authors have declared no conflict of interest.
