## [Peer Review File · Nature Communications]

Reviewers' Comments:

Reviewer #1:

Remarks to the Author:

This is a very good manuscript in the area of NHP models of COVID-19. Its well written and the immunology is of high quality - as would be expected from this corresponding author.

Points to consider:

Abstract is missing the point that 4 of the macaques were infused with plasma (2 with specific and 2 with non-specific plasma).

Line 53-55: There is no supporting data or strong argument provided to make this conclusion "Our data suggest that a vaccine promoting Th1-type Tfh responses that target the S protein may lead to protective immunity" in the abstract.

Authors do not discuss correlation between Tfh responses and antibody development - please comment and include as a caveat.

Lines 84-85: The authors are encouraged to provide a little introduction as to why germinal center Tfh response is critical to vaccine efficacy?

The introduction should end with the questions asked in the present study and a brief summary of what the authors will demonstrate in the further sections.

Line 163: This is very interesting and contradictory to what is already published in great deal in humans. Is the low production of IL-6 the reason for not having a severe disease in macaques despite having been infected with a high viral dose? (PMID: 32475759)

Line 208: These non-SARS-CoV-2 infected animals have not been mentioned prior to this, can the authors comment on why the comparisons are drawn only for the splenic response and not shown for all the previous data?

Lines 223-225: Can the authors comment on how this can impact the vaccine development?
Can the authors comment on cytokines IL-7 and IL-10 produced by polyfunctional cells, as these have been implicated in patients with COVID-19 leading to lung failure, liver, heart and kidney changes (PMID: 31986264)

Line 223-225: "It should be noted, however, that responses to S, N, M were also detected in unexposed animals suggestive of cross-reactive T cells to endemic coronaviruses, as has been reported in humans" Do endemic NHP coronaviruses infect rhesus macaques?

Line 287-288: "Also consistent with reports in infected humans, we observed a strong correlation between neutralization antibody titers and concentrations of anti-RBD IgG antibodies on day 10 (Figure 4E). Together, these data show rapid development of binding and neutralizing antibodies following SARS-CoV-2 infection in the context of mild or absent clinical symptoms."

As authors do not have a comparative group that did not make such antibody response, it would be difficult to make the conclusion that mild form of SARS-CoV-2 infection results in rapid development of binding and neutralizing antibodies.

Figure 1A. Were no lung samples collected in this study? It is not clear in the figure and needs to be made clearer.

Fig 4E: It is interesting that some of the macaques are showing NAb titer of around 1000 at 10dpi.

Again, although the premise of the manuscript is Tfh response and antibody development, authors have not discussed much on the high pseudovirus neutralizing antibody titer observed in these macaques compared to other published paper (Chandrashekhar et al. Science 2020). Discussing it would strengthen the paper.

Figure 1B: Viral load data I+CP (n=2) vs I and I+NP (n=6): although, n=2 is low, there seems to be higher loads (at least one log difference) in this group. Is this because of antibody dependent enhancement of infection. More animals for I+CP would have ruled out this option. However, what if these two macaques are complicating the overall interpretation of the results? Additionally, immune complexes formed by binding antibody (even if neutralizing antibody levels fell below the detection limit) and virus, can also impact the generation of immune responses, as these complexes can be recognized by APCs.

Line 176-178 : "At the apex of the effector response, Ki-67+ CD4 T cells, specifically the Th1 but not the Tfh subset was strongly associated with proliferating CD8 T cells (Fig. 1I). In turn, we observed strong antigen-dependent induction of CD8 T cells evidenced by the association between SARS-CoV-2 vRNA from nasal washes and proliferating (Ki67+) CD8 T cells"

This is direct correlation suggesting that infection is activating the development of CD8 T cell response. Since there is no decrease in viral loads, it would be difficult to consider this observation in the context of correlates of protection resulting in mild disease outcome.

Reviewer #2:

Remarks to the Author:

The authors study the immune response to concurrent intranasal, intratracheal, and ocular SARS-CoV-2 infection in rhesus macaques, with a focus on the CD4+ T cell responses. In this model of mild disease, they find induction of Th1-type Tfh cells (expressing CXCR3) in the mediastinal lymph node along with an IgG-dominated antibody response 7-10 days post infection. They also characterize systemic responses in the blood and the spleen. This is an important study, notably for its characterization of the CD4 T cells at the draining lymph node of the site of infection, but the authors should address the points below to strengthen the conclusions made in the manuscript.

Major comments:

1. The authors argue that Th1-type Tfh cells are likely responsible for the anti-SARS-CoV-2 antibodies that appear by day 7. However, it should be emphasized that this is an association and the necessity of this T cell population for the antibody response cannot be addressed in the current study. Furthermore, the authors should support the claim of Th1-like Tfh cells in the mediastinal lymph node by performing the same intracellular IFN- γ /IL-21 staining they use for splenic T cells (Figure 2C) – but a naïve control will be needed to interpret cytokine induction unless antigen-specific stimulation is used. Although CXCR3 is used to phenotype Th1-polarized cells in peripheral blood and Th1 effector cells in lymph nodes, it is unclear what the combined expression of CXCR3 and CXCR5 would do to follicular localization of presumed Tfh cells and therefore the ability to help B cells.
2. The point of the convalescent plasma inclusion in this study is unclear. It is not mentioned in the abstract nor in any of the figure titles. In Figure 1B, the authors find no effect of convalescent human plasma therapy on viral load, which they argue is due to dilution of the transfused plasma to levels too low to neutralize virus. The authors should show the data supporting this statement. In addition, other studies using similar dosing of convalescent human plasma in patients find that neutralizing titers in the serum increase following plasma transfusion (Duan, 2020, PNAS; Shen, 2020, JAMA). Why do neutralizing antibodies not increase in macaques post-transfusion? It seems this aspect of the study was not well designed or even incorporated into the ultimate findings. It is also unclear whether the study is powered to make any conclusion about convalescent plasma, given only 2 macaques were

infused.

3. In Figures 2C-D and S2E, the authors argue that SARS-CoV-2 infection induces polyfunctional CD4 T cells by showing their production, and sometimes co-production, of multiple cytokines following PMA/ionomycin stimulation. It is essential to have an uninfected macaque for comparison in these figures (like Fig. 3G), as it is unclear whether this is truly infection-induced phenotype. Furthermore, PMA/ionomycin is a supraphysiologic stimulus, so the authors should consider demonstrating cytokine production following antigen-specific stimulation. Finally, it is unclear what the "Unstim" label in Figure S2E refers to, as the corresponding figure legend indicates that this sample is stimulated.

4. Given the observed time course of antibody production in humans after infection, measuring titers only out to day 10 seems inappropriate (ref 26 and 27 do not support the claim made by the authors on line 268 for antibody kinetics). Day 14 is usually the first timepoint considered acceptable for robust antibody detection in humans and would be more consistent with the timing of a Tfh-induced antibody response. Day 7 is quite an early timepoint, and it is possible that the antibody responses at this point may be Tfh-independent. The authors should demonstrate anti-SARS-CoV-2 titers at a later time point, at least 21 days later.

5. Although the authors state that they have identified SARS-CoV-2-specific Tfh cells the data to support this claim is extremely limited (just Fig. 3G). The data presented in fig. 2A does not seem to indicate antigen specificity.

Minor comments:

1. Please cite the AIM assay (line 217).

2. What data are used to construct the t-SNE plots in Figures 1J and S2A? Usually t-SNE plots are used to depict high dimensional data such as scRNA-seq or CyTOF, but it does not seem that such methods were performed.

3 In Figure S3A, it is confusing that the key indicates circles as infected and triangles as infected + infused, but then the graph has circles in the uninfected group.

4. In Figure 3C, it is confusing that there are two blue boxes (FDCs and PD1+ CXCR5+ CD4 T cells) and two green boxes (GC B and PD1- CXCR5+ CD4 T cells).

5. Presumably "P/I" refers to pma/ionomycin in Fig. 3G? This information should be added to the figure legend.

6. Recent work has highlighted that SLAM is a Tfh-associated rather than Th1-associated molecule. Similarly, the authors state "effector molecule CX3CR1, a marker potentially for newly generated memory CD4 T cell subsets", but is used in this study (and more typically in most studies) to identify Th1-skewed cells. The authors should support their rationale for multiple aspects of defining populations throughout the study.

Reviewer #3:

Remarks to the Author:

In the manuscript "SARS-COV-2 infection induces robust germinal center CD4 T follicular helper cell responses in rhesus macaques", the authors report that SARS-CoV-2 infection results in transient accumulation of activated and proliferating Tfh cells of Th1 phenotype in the blood and MLN.

The infusion of convalescent human plasma is a great control, unfortunately the neutralization titer fell below detectable limits. Why didn't the authors give more plasma to get a better response? I would guess the reason is just the availability of sufficient amounts of plasma and the known overall low neutralization titers in patie4nts. With all the concepts going on to use human plasma to treat patients it had been so interesting to see whether higher doses had resulted in blunting viral load.

I wonder why the infected animals did not show any clinical symptoms of illness upon infection?

The presence of PD-1 CXCR5 expressing cells is a good indicator for ongoing GC reactions. I wonder how a B cell staining looks like in these animals. CD95, Peanut agglutinin or GL-7 in combination with CD38? Germinal center are easy to detect in immunofluorescent analysis of frozen tissue sections. Were any of these analysis done?

It is interesting to see that responses to S, N, M proteins were also detected in non infected animals. As the authors state this was also observed in humans and discussed as cross-reactive T cells to endemic coronaviruses. I wonder how big the chance is for those colony-bred animals housed in an animal research center to get contact with other coronaviruses.

The data presented convincingly show that S and N specific CD4 Tfh cells are induced upon SARS-CoV-2 infection

In figure 3C the authors show that 23.2% of B cells (CD20+) express Bcl6. CD95 is also present in the staining cocktail Can the authors gate CD95+ B cells (=germinal center B cells) and show that frequency of Bcl6 expression is higher in the CD95+ B cell compartment. This would help to nail the point of GC induction (though without information about specificity for Sars-CoV-2).

Figure 4 would benefit from IgG isotype data. Do these animals display IgG1 and IgG2? This would help the understanding whether the immune response is exclusively Th1 or alsoTh2.

Overall I think. This study would highly benefit from showing the presence of Germinal center reactions e.g. in the MLN by any type of microscopy (preferentially immunofluorescence).

Germinal center reactions are the source of mutated antibodies. Is there any evidence of hypermutation of antibodies?

Point-by-point response to reviewers comments

Reviewer #1

This is a very good manuscript in the area of NHP models of COVID-19. Its well written and the immunology is of high quality - as would be expected from this corresponding author.

Point # 1. Abstract is missing the point that 4 of the macaques were infused with plasma (2 with specific and 2 with non-specific plasma).

Response: As requested, we have now referenced the plasma infusion in the abstract (Page 2, Ln 53-55).

Point # 2. Line 53-55: There is no supporting data or strong argument provided to make this conclusion “Our data suggest that a vaccine promoting Th1-type Tfh responses that target the S protein may lead to protective immunity” in the abstract.

Response: The reviewer’s point is well taken. We have removed this conclusion from the Abstract.

Point # 3. Authors do not discuss correlation between Tfh responses and antibody development - please comment and include as a caveat.

Response: As requested, we have now included data showing that the proportion of T_h1 T_{fh} cells at day 7 correlates with antibody responses against RBD at day 10 (Figure 7C, last panel on the right, Page 19, Ln 500-503).

Point # 4. Lines 84-85: The authors are encouraged to provide a little introduction as to why germinal center Tfh response is critical to vaccine efficacy?

Response: As suggested, we have now provided an introduction as to the importance of Tfh cells in vaccine efficacy (Page 3, Ln 87-91)

Point # 5. The introduction should end with the questions asked in the present study and a brief summary of what the authors will demonstrate in the further sections.

Response: We have now incorporated these suggestions – the introduction ends with questions asked in the present study and a brief summary of the main findings (Page 4, Ln 106-110).

Point # 6. This is very interesting and contradictory to what is already published in great deal in humans. Is the low production of IL-6 the reason for not having a severe disease in macaques despite having been infected with a high viral dose? (PMID: 32475759)

Response: The reviewer raises an important point. We have now indicated the lack of systemic increase in IL-6 as a possible factor underlying mild/asymptomatic disease (Page 6, Ln 170-172).

Point # 7. Line 208: These non-SARS-CoV-2 infected animals have not been mentioned prior to this, can the authors comment on why the comparisons are drawn only for the splenic response and not shown for all the previous data?

Response: We have now clarified that SARS-CoV-2 unexposed tissue was obtained from opportunistic necropsies (Page 8, Ln 222-224)

Point # 8. Lines 223-225: Can the authors comment on how this can impact the vaccine development?

Response: We have now included a point in the results about how pre-existing T cell immunity could impact vaccine development (Page 8, Ln 225-228).

Point # 9. Can the authors comment on cytokines IL-7 and IL-10 produced by polyfunctional cells, as these have been implicated in patients with COVID-19 leading to lung failure, liver, heart and kidney changes (PMID: 31986264)

Response: IL-7 and IL-10 have been reported to be higher in ICU compared to non-ICU patients. However, these cytokines are produced by a range of cells and are not linked to polyfunctional T cells.

Point # 10. Line 223-225: “It should be noted, however, that responses to S, N, M were also detected in unexposed animals suggestive of cross-reactive T cells to endemic coronaviruses, as has been reported in humans” Do endemic NHP coronaviruses infect rhesus macaques?

Response: We have now referenced an article demonstrating natural infection of rhesus macaques with HCoV-NL63 (Page 8, Ln 224-225).

Point # 11. Line 287-288: “Also consistent with reports in infected humans, we observed a strong correlation between neutralization antibody titers and concentrations of anti-RBD IgG antibodies on day 10 (Figure 4E). Together, these data show rapid development of binding and neutralizing antibodies following SARS-CoV-2 infection in the context of mild or absent clinical symptoms.” As authors do not have a comparative group that did not make such antibody response, it would be difficult to make the conclusion that mild form of SARS-CoV-2 infection results in rapid development of binding and neutralizing antibodies.

Response: We have now clarified the text to emphasize development of antibody responses (Page 12, Ln 316-319).

Point # 12. Figure 1A. Were no lung samples collected in this study? It is not clear in the figure and needs to be made clearer.

Response: We have now indicated collection of lung samples in Figure 1A.

Point # 13. Fig 4E: It is interesting that some of the macaques are showing NAb titer of around 1000 at 10dpi. Again, although the premise of the manuscript is T_{fh} response and antibody development, authors have not discussed much on the high pseudovirus neutralizing antibody titer observed in these macaques compared to other published paper (Chandrashekhar et al. Science 2020). Discussing it would strengthen the paper.

Response: The reviewer raises a good point; we have now included a discussion of this observation in the results (Page 12, Ln 312-314)

Point # 14. Figure 1B: Viral load data I+CP (n=2) vs I and I+NP (n=6): although, n=2 is low, there seems to be higher loads (at least one log difference) in this group. Is this because of antibody dependent enhancement of infection. More animals for I+CP would have ruled out this option. However, what if these two macaques are complicating the overall interpretation of the results? Additionally, immune complexes formed by binding antibody (even if neutralizing antibody levels fell below the detection limit) and virus, can also impact the generation of immune responses, as these complexes can be recognized by APCs.

Response: We agree that including more animals in the I+CP group would have allowed us to address these important questions and now explain this caveat in the discussion section (Page 14, Ln 360- 370).

Point # 15. Line 176-178: “At the apex of the effector response, Ki-67+ CD4 T cells, specifically the Th1 but not the Tfh subset was strongly associated with proliferating CD8 T cells (Fig. 1I). In turn, we observed strong antigen-dependent induction of CD8 T cells evidenced by the association between SARS-CoV-2 vRNA from nasal washes and proliferating (Ki67+) CD8 T

cells” This is direct correlation suggesting that infection is activating the development of CD8 T cell response. Since there is no decrease in viral loads, it would be difficult to consider this observation in the context of correlates of protection resulting in mild disease outcome.

Response: The reviewer’s point about antigen-driven expansion of CD8 T cells is an important one. We have included data showing that such a correlation also exists for lung Granzyme B and PD-1+ CD8 T cell responses (Figure 4C, Page 9, Ln 248- 250)

Reviewer #2

The authors study the immune response to concurrent intranasal, intratracheal, and ocular SARS-CoV-2 infection in rhesus macaques, with a focus on the CD4+ T cell responses. In this model of mild disease, they find induction of T_h1-type Tfh cells (expressing CXCR3) in the mediastinal lymph node along with an IgG-dominated antibody response 7-10 days post infection. They also characterize systemic responses in the blood and the spleen. This is an important study, notably for its characterization of the CD4 T cells at the draining lymph node of the site of infection, but the authors should address the points below to strengthen the conclusions made in the manuscript.

Point # 1. 1. The authors argue that Th1-type Tfh cells are likely responsible for the anti-SARS-CoV-2 antibodies that appear by day 7. However, it should be emphasized that this is an association and the necessity of this T cell population for the antibody response cannot be addressed in the current study.

Response: We agree with the reviewer and have modified the text to emphasize this point (Page 15, Ln 387-389).

Point # 2. Furthermore, the authors should support the claim of T_h1-like Tfh cells in the mediastinal lymph node by performing the same intracellular IFN-γ/IL-21 staining they use for splenic T cells (Figure 2C) – but a naive control will be needed to interpret cytokine induction unless antigen-specific stimulation is used.

Response: Unfortunately, due ramping down of research activities at the primate center, mediastinal lymph nodes from uninfected controls were not attainable. We now include data showing enrichment of CD40L+ IFNG+ cells within GC Tfh compartment. These functional data demonstrate the induction of Th1-Tfh cells following SARS-CoV-2 infection (Figure S5D, Page 10, Ln 274-277)

Point # 3. Although CXCR3 is used to phenotype T_h1-polarized cells in peripheral blood and Th1 effector cells in lymph nodes, it is unclear what the combined expression of CXCR3 and CXCR5 would do to follicular localization of presumed Tfh cells and therefore the ability to help B cells.

Response: As requested, we have now included extensive phenotyping of CXCR3+ and CXCR3- cells for a number of markers including CXCR5 and CCR7 which control follicular localization (Figure 5G, Page 11, Ln 279- 290)

Point # 4. The point of the convalescent plasma inclusion in this study is unclear. It is not mentioned in the abstract nor in any of the figure titles. In Figure 1B, the authors find no effect of convalescent human plasma therapy on viral load, which they argue is due to dilution of the transfused plasma to levels too low to neutralize virus. The authors should show the data supporting this statement. In addition, other studies using similar dosing of convalescent human plasma in patients find that neutralizing titers in the serum increase following plasma transfusion

(Duan, 2020, PNAS; Shen, 2020, JAMA). Why do neutralizing antibodies not increase in macaques post-transfusion? It seems this aspect of the study was not well designed or even incorporated into the ultimate findings. It is also unclear whether the study is powered to make any conclusion about convalescent plasma, given only 2 macaques were infused.

Response: As requested, we have now referenced plasma infusion in the Abstract, Results, and the Discussion and included data showing human binding and neutralizing antibody in infused animals (Figure 1). We have also included a caveat of our study design of CP (Discussion, Page 14, Ln 360 - 370).

Point # 5. In Figures 2C-D and S2E, the authors argue that SARS-CoV-2 infection induces polyfunctional CD4 T cells by showing their production, and sometimes co-production, of multiple cytokines following PMA/ionomycin stimulation. It is essential to have an uninfected macaque for comparison in these figures (like Fig. 3G), as it is unclear whether this is truly infection-induced phenotype.

Response: We have now included polyfunctionality plots from contemporaneously assayed uninfected controls (Figure 3, Page 9, Ln 233-234)

Point # 6. Furthermore, PMA/ionomycin is a supraphysiologic stimulus, so the authors should consider demonstrating cytokine production following antigen-specific stimulation.

Response: We now have included data from splenocytes stimulated with SARS-CoV-2 peptide pools showing cytokine responses. (Supplementary Figure S3C-D, Page 9, Ln 234-236)

Point # 7. Finally, it is unclear what the “Unstim” label in Figure S2E refers to, as the corresponding figure legend indicates that this sample is stimulated.

Response: We have now clarified in the figure and figure legend that the “unstimulated” label is an overlay of cytokine responses in unstimulated cells

Point # 8. Given the observed time course of antibody production in humans after infection, measuring titers only out to day 10 seems inappropriate (ref 26 and 27 do not support the claim made by the authors on line 268 for antibody kinetics). Day 14 is usually the first timepoint considered acceptable for robust antibody detection in humans and would be more consistent with the timing of a Tfh-induced antibody response. *Day 7 is quite an early timepoint, and it is possible that the antibody responses at this point may be Tfh-independent. The authors should demonstrate anti-SARS-CoV-2 titers at a later time point, at least 21 days later.*

Response: It is true that the early response may be partly Tfh-independent. Unfortunately, the macaques in our study were sacrificed before day 21 so we cannot show antiviral antibody titers at this time point. However, the delayed appearance of serum IgA antibodies compared to IgM and IgG, and our new results showing that some of these animals also had IgG2, IgG3 or IgG4 antibodies on day 10 (Figure 7B) suggest that class-switching, a predominantly Tfh-dependent reaction, was occurring.

Point # 9. Although the authors state that they have identified SARS-CoV-2-specific Tfh cells the data to support this claim is extremely limited (just Fig. 3G). The data presented in fig. 2A does not seem to indicate antigen specificity.

Response: The data presented Figures 3A-B, Figure 3E-F, Figure 5H, and Supplementary Figure S3C-D demonstrate SARS-CoV-2 specific Tfh cells.

Minor comments:

1. Please cite the AIM assay (line 217).

Response: We have now included a citation of the AIM assay (Page 8, Ln 216)

2. What data are used to construct the t-SNE plots in Figures 1J and S2A? Usually t-SNE plots are used to depict high dimensional data such as scRNA-seq or CyTOF, but it does not seem that such methods were performed.

Response: We have now indicated that t-SNE plots were constructed using flow cytometry data in the results and the figure legends (Page 7, Ln 185; Page 17, Ln 450).

3 In Figure S3A, it is confusing that the key indicates circles as infected and triangles as infected + infused, but then the graph has circles in the uninfected group.

Response: We thank the reviewer for catching this. We have modified symbols in the uninfected group to squares to improve clarity.

4. In Figure 3C, it is confusing that there are two blue boxes (FDCs and PD1+ CXCR5+ CD4 T cells) and two green boxes (GC B and PD1- CXCR5+ CD4 T cells).

Response: We have now represented PD-1+ and PD-1 - CXCR5+ CD4 T cells in colors distinct from FDCs and GC B cells (Figure 5B).

5. Presumably “P/I” refers to pma/ionomycin in Fig. 3G? This information should be added to the figure legend.

Response: The Figure legend has been updated with this information (Page 18, Ln 462)

6. Recent work has highlighted that SLAM is a Tfh-associated rather than Th1-associated molecule. Similarly, the authors state “effector molecule CX3CR1, a marker potentially for newly generated memory CD4 T cell subsets”, but is used in this study (and more typically in most studies) to identify Th1-skewed cells. The authors should support their rationale for multiple aspects of defining populations throughout the study.

Response: We have now clarified the use of markers in the results section (Page 7, Ln 188-193)

Reviewer #3

In the manuscript “SARS-COV-2 infection induces robust germinal center CD4 T follicular helper cell responses in rhesus macaques”, the authors report that SARS-CoV-2 infection results in transient accumulation of activated and proliferating Tfh cells of Th1 phenotype in the blood and MLN.

Point # 1. The infusion of convalescent human plasma is a great control, unfortunately the neutralization titer fell below detectable limits. Why didn't the authors give more plasma to get a better response? I would guess the reason is just the availability of sufficient amounts of plasma and the known overall low neutralization titers in patients. With all the concepts going on to use human plasma to treat patients it had been so interesting to see whether higher doses had resulted in blunting viral load.

Response: The reviewer raises some very important points. We have now included these points in the discussion (Page 14, Ln 360-370)

Point # 2. I wonder why the infected animals did not show any clinical symptoms of illness upon infection?

Response: We now discuss the transient innate immune response and lack of discernible increase in systemic IL-6 as factors underlying mild clinical disease observed (Page 6, Ln 168-172)

Point # 3. The presence of PD-1 CXCR5 expressing cells is a good indicator for ongoing GC reactions. I wonder how a B cell staining looks like in these animals. CD95, Peanut agglutinin or GL-7 in combination with CD38? Germinal center are easy to detect in immunofluorescent analysis of frozen tissue sections. Were any of these analysis done?

Response: As requested, we have performed both H&E and immunofluorescence which now complement quantitative flow cytometry data showing germinal center responses in the mediastinal lymph nodes (Figure S4C, Figure 5A, Page 10, Ln 254-258).

Point # 4. It is interesting to see that responses to S, N, M proteins were also detected in non-infected animals. As the authors state this was also observed in humans and discussed as cross-reactive T cells to endemic coronaviruses. I wonder how big the chance is for those colony-bred animals housed in an animal research center to get contact with other coronaviruses.

Response: We have now referenced an article showing natural infection of rhesus macaques with HCoV-NL63 (Page 8, Ln 224- 225).

Point # 5. The data presented convincingly show that S and N specific CD4 Tfh cells are induced upon SARS-CoV-2 infection

Response: We are appreciative of the reviewer's positive feedback.

Point # 6. In figure 3C the authors show that 23.2% of B cells (CD20+) express Bcl6. CD95 is also present in the staining cocktail Can the authors gate CD95+ B cells (=germinal center B cells) and show that frequency of Bcl6 expression is higher in the CD95+ B cell compartment. This would help to nail the point of GC induction (though without information about specificity for Sars-CoV-2).

Response: As requested, we have now gated on GC B cells according to the reviewer's suggestion (Figure 5B, D)

Point # 7. Figure 4 would benefit from IgG isotype data. Do these animals display IgG1 and IgG2? This would help the understanding whether the immune response is exclusively Th1 or alsoTh2.

Response: As requested, we have added graphs that illustrate the IgG subclass responses in our animals (Figure 7B). These new results show that the IgG response to SARS antigens is predominantly IgG1. This finding supports the induction of a T_h1 type response because we have previously shown that T_h1 responses in macaques are associated with dominant IgG1 responses whereas T_h2 responses are characterized by the induction of IgG4.

Point # 8. Overall, I think. This study would highly benefit from showing the presence of Germinal center reactions e.g. in the MLN by any type of microscopy (preferentially immunofluorescence).

Response As requested and stated in Point #3, we have performed both H&E and immunofluorescence which now complement quantitative flow cytometry data showing germinal center responses in the mediastinal lymph nodes (Figure S4C, Figure 5A, Page 10, Ln 254-258).

Point # 9. Germinal center reactions are the source of mutated antibodies. Is there any evidence of hypermutation of antibodies?

Response. This a very important point. Unfortunately, we are unable to address in the present studies.

Reviewers' Comments:

Reviewer #1:

Remarks to the Author:

The authors have made all changes necessitated by the first round of review process. As such, I have no concerns and I think this is a very good paper.

Reviewer #2:

Remarks to the Author:

We commend the authors for their revised manuscript, which is greatly improved by their addition of more flow characterization, including antigen-specific cytokine production by T cell subsets, immunofluorescence images, and discussion of caveats of transfusion experiments. The authors should address the points below to further strengthen the conclusions made in the manuscript.

Major comments:

1. In Fig 1, the authors demonstrate that transfusion of convalescent plasma leads to increase in SARS-CoV-2-specific human IgG in serum of recipients. However, the same serum from recipients does not have detectable pseudovirus neutralization titer, compared to the high neutralization titer of the transferred plasma. If there is remaining convalescent plasma, could the authors measure levels of SARS-CoV-2-specific antibodies (as in Fig 1B)? It would be helpful to get a sense of the concentration of antibodies (e.g. ug/ml or mg/ml?) that is required to neutralize virus.
2. The authors argue that "the appearance of antiviral IgG antibodies by day 7 with delayed induction of IgA responses suggests that early class-switching occurs after SARS-CoV-2 infection and is likely promoted by Th1-type Tfh cells" (lines 319-321). This is not a strong point as IgA can be induced in a Tfh-independent manner (Bunker et al Immunity 2015, Zhang et al Science Immunology 2020, Bai et al J. Immunology 2020). Furthermore, IFN-g as produced by Th1-type Tfh cells is not a switch factor for IgA.

Minor comments:

1. The in-text descriptions of Fig 5 do not match the figure content, e.g. there is no Fig 5H.
2. In Fig 3D, the authors show that T cells from uninfected animals show similar polyfunctionality as infected animals. Therefore, the authors should explicitly write in the text that the polyfunctionality observed is not specific to SARS-CoV-2 infection.

Reviewer #3:

Remarks to the Author:

I am very, very happy with the revised manuscript. All my points (except #9 which I agree is too challenging) were addressed by the authors and the revised version represents a solid study. The immunofluorescent analysis is exceptional. All additional data presented by the authors fully support the authors conclusion. Statistical analysis is appropriate everywhere.

I have no further comments or concerns.

Point-by-point response to reviewers comments

Reviewer #1

The authors have made all changes necessitated by the first round of review process. As such, I have no concerns and I think this is a very good paper.

Response: We appreciate the reviewer's recognition of our effort to revise the manuscript.

Reviewer #2

We commend the authors for their revised manuscript, which is greatly improved by their addition of more flow characterization, including antigen-specific cytokine production by T cell subsets, immunofluorescence images, and discussion of caveats of transfusion experiments. The authors should address the points below to further strengthen the conclusions made in the manuscript.

Major comments:

1. In Fig 1, the authors demonstrate that transfusion of convalescent plasma leads to increase in SARS-CoV-2-specific human IgG in serum of recipients. However, the same serum from recipients does not have detectable pseudovirus neutralization titer, compared to the high neutralization titer of the transferred plasma. If there is remaining convalescent plasma, could the authors measure levels of SARS-CoV-2-specific antibodies (as in Fig 1B)? It would be helpful to get a sense of the concentration of antibodies (e.g., ug/ml or mg/ml?) that is required to neutralize virus.

Response: We have now included data showing concentrations of S1, S2, and N antibodies (IgM and IgG) in pooled convalescent plasma (**Figure 1C**).

2. The authors argue that “the appearance of antiviral IgG antibodies by day 7 with delayed induction of IgA responses suggests that early class-switching occurs after SARS-CoV-2 infection and is likely promoted by Th1-type Tfh cells” (lines 319-321). This is not a strong point as IgA can be induced in a Tfh-independent manner (Bunker et al Immunity 2015, Zhang et al Science Immunology 2020, Bai et al J. Immunology 2020). Furthermore, IFN-g as produced by Th1-type Tfh cells is not a switch factor for IgA.

Response: The reviewer's point is well taken. We have clarified our statement to state that IgA can be induced in a Tfh-independent manner (**Figure 1C, Ln 326 – 327**).

Minor comments:

1. The in-text descriptions of Fig 5 do not match the figure content, e.g., there is no Fig 5H.

Response: We thank the reviewer for pointing this out. We have now corrected this error.

2. In Fig 3D, the authors show that T cells from uninfected animals show similar polyfunctionality as infected animals. Therefore, the authors should explicitly write in the text that the polyfunctionality observed is not specific to SARS-CoV-2 infection.

Response: We have now included this statement (**Ln 237 – 238**).

Reviewer #3

I am very, very happy with the revised manuscript. All my points (except #9 which I agree is too challenging) were addressed by the authors and the revised version represents a solid study. The immunofluorescent analysis is exceptional. All additional data presented by the authors fully support the authors conclusion. Statistical analysis is appropriate everywhere. I have no further comments or concerns.

Response: We appreciate the reviewer's recognition of our effort to revise the manuscript.